# Influence of Paratuberculosis Vaccination on the Local Immune Response in Experimentally Infected Calves: An Immunohistochemical Analysis

**DOI:** 10.3390/ani15131841

**Published:** 2025-06-22

**Authors:** David Zapico, José Espinosa, María Muñoz, Luis Ernesto Reyes, Julio Benavides, Juan Francisco García Marín, Miguel Fernández

**Affiliations:** 1Departamento de Sanidad Animal, Facultad de Veterinaria, Universidad de León, C/ Profesor Pedro Cármenes s/n, E-24071 León, Spain; jespic@unileon.es (J.E.); mmunf@unileon.es (M.M.); lreyesvet@gmail.com (L.E.R.); jfgarm@unileon.es (J.F.G.M.); m.fernandez@unileon.es (M.F.); 2Departamento de Sanidad Animal, Instituto de Ganadería de Montaña (CSIC-ULE), E-24346 Grulleros, Spain; julio.benavides@csic.es

**Keywords:** paratuberculosis, lesion, granuloma, cattle, vaccination, TLR, IFN-γ, macrophage, M1, M2

## Abstract

Vaccination is the most cost-effective method to control paratuberculosis in dairy cattle, but its effects on the local immune response in the intestine are not well understood. This study evaluated the local innate and cell-mediated response in calves vaccinated with Silirum^®^ and subsequently infected with paratuberculosis. Samples were taken from injection-site granuloma, scapular and mesenteric lymph nodes, and the intestine. Results show that vaccinated calves with focal forms controlling the infection displayed a robust innate response at the intestine, with a strong cell-mediated immunity in the granulomas. An upregulated innate and pro-inflammatory response was also observed in the site of the injection. In contrast, animals with progression of the disease with multifocal or diffuse lesions, both vaccinated and not vaccinated, showed weak intestinal innate immunity and an anti-inflammatory response in the granulomas. In summary, vaccination with Silirum^®^ enhances the immune response in the intestine, potentially contributing to protection against paratuberculosis.

## 1. Introduction

*Mycobacterium avium* subspecies *paratuberculosis* (MAP) is the causative agent of Johne’s disease [1], a chronic enteritis that causes important economic losses in the livestock industry [2]. The implementation of vaccines represents a useful tool to control the clinical disease in dairy herds [3,4,5] but does not provide total protection against the infection [6]. Silirum^®^ (C. Z. Veterinaria S.A., Porriño, Pontevedra, España) is a commercial vaccine containing heat-inactivated MAP strain 316F formulated in an oil-based adjuvant. Silirum^®^ induces a strong, long-lasting Th1/Th2 peripheral response in calves experimentally infected with paratuberculosis [7]. Nonetheless, the immunological events that take place in the blood may not fully represent those at the site of the infection [8,9,10], which have been less explored. The study of the local immune response is critical for understanding the host–pathogen interaction and the mechanisms of disease progression, which is essential for the design of effective immunoprophylactic strategies.

Toll-like receptors (TLRs) are a family of pattern recognition receptors (PRRs) involved in microbial detection by innate immune cells [11]. Several of these receptors, including toll-like receptor (TLR)-1, 2, 4 and 9, have been associated with host susceptibility/resistance to paratuberculosis in cattle [12], but there are fewer studies about their expression at the infection site [9,13]. TLRs participate not only in the innate detection of invading pathogens, but also in the onset of interferon (IFN)-γ dependent adaptive immunity against mycobacteria [14].

IFN-γ promotes the activation of macrophages for the killing of intracellular bacteria [15,16], playing a key role in the pathogenesis of paratuberculosis [17,18,19,20,21]. In fact, macrophages can exist in two distinct states: M1 or classically activated, which display antimicrobial activity, and M2 or alternatively activated, which have immunoregulatory functions [22]. IFN-γ production and M1-polarized macrophages have been associated with the containment of the infection within the granulomas, while anti-inflammatory M2 macrophages abound in the intestine of heavily infected cows with lower levels of IFN-γ [21,23,24]. These macrophage subsets can be distinguished by the expression of different markers, such as inducible nitric oxide synthase (iNOS) for M1 macrophages and cluster of differentiation (CD)-204 receptor for M2 macrophages, among others [25,26].

Recently, intradermal vaccination against MAP was found to result in an enhanced pro-inflammatory response at the intestine of rabbit and sheep infected with MAP [27,28], though these results have not yet been reported for cattle.

This study aimed to evaluate the effects of parenteral vaccination with Silirum^®^ on the local immune response through the immunohistochemical analysis of TLR1, TLR2, TLR4, TLR9, IFN-γ, iNOS and CD204 in the intestine of calves experimentally infected with paratuberculosis.

## 2. Materials and Methods

### 2.1. Ethical Approval

Archived formalin-fixed embedded tissue (jejunum, ileum, ileocecal valve, scapular and mesenteric lymph nodes, and injection-site granuloma) samples of 16 Holstein calves from a previous study [7] were used for this study.

### 2.2. Tissue Samples and Classification of the Lesions

Selected samples were categorized as focal, multifocal and diffuse paucibacillary (lymphocytic), as previously described [29], and came from the following groups: (i) vaccinated and infected (*n* = 6): 4 calves with focal lesions and 2 with multifocal lesions; (ii) non-vaccinated and infected (*n* = 6): 2 calves with focal lesions, 3 with multifocal lesions and 1 with diffuse lymphocytic lesions; (iii) non-vaccinated and not infected (*n* = 2) and (iv) vaccinated and not infected (*n* = 2).

Focal lesions were composed of small and well-demarcated granulomas located exclusively in the interfollicular areas of the Peyer’s patches and lymph nodes. Multifocal lesions were characterized by small granulomas located not only in the lymphoid tissue (as in the focal lesions) but also in the intestinal lamina propria. Diffuse paucibacillary or lymphocytic lesions consisted in a widespread lymphocytic infiltrate, with some granulomas and Langhans giant cells scattered among the lymphocytes.

### 2.3. Immunohistochemistry

Primary TLR1 (GTX47794), TLR2 (MA1-40080), TLR4 (ORB11489), TLR9 (GTX31295), IFN-γ (MCA2112), iNOS (MA3-030) and CD204 (KAL-KT022) antibodies with verified customer reviews of reactivity in sections of formalin-fixed tissues were used (Appendix A). Heat-mediated antigen retrieval was performed with the PT Link^®^ (Dako-Agilent^®^ technologies, Santa Clara, CA, USA) system, using either pH 6.0 target retrieval solution for TLR1, TLR2, iNOS and CD204, or pH 9.0 solution for TLR4, TLR9 and IFN-γ, for 20 min at 95 °C. Immunohistochemistry was carried out as described elsewhere [30]. Briefly, sections were immersed into a 3% H_2_O_2_ in methanol solution for 30 min at room temperature to block endogenous tissue peroxidase. Samples were then incubated overnight with the primary antibody in a humidified chamber at 4 °C. Immunolabeling was performed using the EnVision System^®^ (Dako-Agilent technologies, Glostrup, Denmark). Slides were incubated with appropriate anti-mouse or anti-rabbit secondary antibody for 40 min at room temperature. Antibody localization was determined using 3,3-diaminobenzidine (Dako-Agilent technologies, Glostrup, Denmark) as chromogenic substrate and slides were then counterstained with hematoxylin. Appropriate species- and isotype-matched immunoglobulins were used as negative controls. These included sections with an isotype control for the primary antibody and the omission of the primary antibody.

The specificity of TLR1, TLR2, TLR4 and TLR9 primary antibodies on the bovine species was confirmed by Western blot analysis, as previously described [13].

### 2.4. Evaluation of the Immunolabeling

For TLR1, TLR2, TLR4 and IFN-γ antibodies, samples were scored according to the number of positively immunolabeled cells. Due to the heterogeneity in the nature and distribution of the immunolabeled cells, a differential cell count was carried out in 30 randomly chosen fields from each of the following locations: lamina propria (LP), gut-associated lymphoid tissue (GALT), mesenteric lymph node (MLN), scapular LN (SLN) and injection-site subcutaneous granuloma (ISG) of each sample included in the study, at 400× (Nikon^®^ Eclipse Ci microscope with Digital MD-E3-6-3 digital camera, Nikon, Minato ku, Japan). The mean value on each location (ISG, SLN, LP, GALT, MLN) was independently obtained for each lesion category (control, focal, multifocal, diffuse lymphocytic) according to the vaccination status (vaccinated/not vaccinated). The type of cells that were immunolabeled was assessed according to morphological features and, additionally, their distribution was evaluated in relation to the granulomas.

For iNOS and CD204 antibodies, only those cells with a clear macrophage morphology and which formed part of the granulomas were considered. As the macrophages forming the MAP-associated lesions showed differences in the immunolabeling intensity and in the number of labeled cells, samples were semi-quantitatively scored using a complete histological score (H-score), which considers both the percentage of labeled cells and the immunolabeling intensity [31]. The same analysis was performed for TLR9 antibody as its immunolabeling was virtually restricted to the granulomas. Briefly, five randomly chosen fields from the intestine containing granulomas were evaluated at 630× in a total of 5 different tissue slides (25 fields) for each type of lesion according to vaccination status. This analysis was performed in the ISG of vaccinated cattle considering the type of lesion present in the intestine. Macrophages forming the granulomas were subjectively classified into 4 different categories consistent with the presence of negative, mild, moderate or intense immunolabeling. H-score was then calculated by adding the percentage of macrophages of each category multiplied by the immunolabeling intensity score (0, negative; 1, mild; 2, moderate and 3, intense). H-score between 0 and 300 was obtained for each type of lesion according to the vaccination status, considering the mean value of the 25 areas evaluated.

Evaluation of the cell counting and histological scoring (H-score) was performed independently and in duplicate (D.Z., training pathologist, and J.E, boarded pathologist) and discordant results were discussed in a multi-headed microscope by both pathologists to reach consensus.

### 2.5. Statistical Analysis

To explore the effect of different variables on the number of positively immunolabelled cells, generalized linear models (GLMs) with a negative binomial distribution were used. Specifically, the number of TLR1, TLR2, TLR4, TLR9, IFN-γ, iNOS, and CD204 positive cells evaluated in the intestine, ISG, and SLN were the response variables. The explanatory variables included infection status (IS) (infected/non-infected), vaccination status (VS) (vaccinated/non-vaccinated), type of lesion (TL) (non-lesion, focal, multifocal, and diffuse paucibacillary), and intestinal area (IA) (LP, GALT, and MLN). The intestinal area was excluded as an explanatory variable when analyzing the different markers in the SLN or ISG.

Previously, in all cases, the special underlying assumptions of GLMs (linear relationships between the dependent and independent variables) were confirmed. Considering the characteristics of our data set and in order to apply the most informative method, we used an information-theoretic approach [32,33] based on the Akaike information criterion corrected for small sample sizes (AICc) [34]. The dredge, get.models and model.sel functions included in R package ‘MuMIn’ were used to construct a set of candidate models with all of the possible combinations of predictive variables. Model selection identified in our analyses the most parsimonious model (lowest AICc) [34] from the possible sub-sets, which ranged from the null model to a model with explanatory variables and their two-order interactions. Models with Δi < 2 units have substantial support for explaining the observed variability in the variable of interest [34]. Subsequently, we estimated the Akaike weight (Wi) and the percentage of explained variability (*R*^2^) for each selected model [34]. Specific GLM requirements (e.g., homoscedasticity, normality, multicollinearity and overdispersion) were assessed using diagnostic graphics (the R packages ‘mctest’ and ‘car’) before model interpretation [35]. Finally, to observe the differences between groups within the relevant variables included in the final fitted model, we used the Tukey’s honestly significant difference adjustment for the whole pairwise comparisons using the glht.function with the “multcomp” package in R. *p*-values of less than 0.05 were considered statistically significant.

All statistical analyses were performed with the R software version 4.4.1 (R Core Team, Vienna, Austria).

## 3. Results

### 3.1. Distribution of the Immunolabeled Cells

TLR1, TLR2, TLR4, TLR9, iNOS, and CD204 immunolabeling was observed in macrophages, while IFN-γ labeling was detected predominantly in lymphocytes. Nevertheless, TLR2, IFN-γ and iNOS positivity was also found in neutrophils. Paneth cells in the crypts of Lieberkühn showed positive immunolabeling for TLR4 and iNOS.

In control calves, the distribution of the positively immunolabeled cells was independent of the vaccination status (Figure 1A,D,G,J and Figure 2A,E,I,M). TLR1, TLR2, TLR4, TLR9, iNOS, and IFN-γ immunolabeling was restricted to the cell cytoplasm, whereas CD204 exhibited membranous labeling. Variable numbers of labeled macrophages for TLR1 (Figure 1D and Figure 2E), TLR2, TLR4, TLR9, iNOS, CD204 and lymphocytes immunolabeled for IFN-γ, were scattered through the LP. Individual Paneth cells at crypts showed positive immunolabeling for iNOS and, to a lesser extent, TLR4 in the cytoplasm. Macrophages immunolabeled for TLR2 (Figure 1G and Figure 2I) and lymphocytes labeled for IFN-γ (Figure 2A) clustered at the dome of the Peyer’s patches, while those in the interfollicular region were labeled for TLR4 (Figure 1J). Fewer TLR1+, TLR2+, TLR4+, TLR9+, iNOS+, CD204+ and IFN-γ+ cells were also detected in the GALT, and always in lower numbers. The immunolabeled cells showed a similar distribution at the SLN and MLN. Thus, isolated macrophages in the medullary cords labeled for iNOS, TLR1, TLR2, TLR4 and TLR9, while those at the sinus showed positive immunolabeling for CD204, with scarce TLR4 and TLR9 immunolabeling. Lymphocytes immunolabeled for IFN-γ were observed in the medullary cords and sinus but frequently gathered at the corticomedullary junction (Figure 1A). Scarce positive cells for the different antibodies were present in the cortex.

In samples with focal lesions from both vaccinated (Figure 1B,E,H,K) and not vaccinated (Figure 2B,F,J,N) calves, the positively immunolabeled cells situated outside of the granulomas followed a similar distribution to the controls. Nonetheless, in sections from both vaccinated and not vaccinated cattle, moderate numbers of lymphocytes labeled for IFN-γ were seen in close relationship to the granulomas present in the GALT (Figure 1B insert and Figure 2B) and intestinal LN, which showed diffuse cytoplasmic immunolabeling for TLR9 antibody (Figure 6B). Aside from TLR9 immunolabeling, the macrophages forming the core of the focal lesions from calves vaccinated with Silirum® showed an inconsistent labeling for iNOS in the cytoplasm (Figure 6D), with scarce cytoplasmic CD204 immunolabeling of those present in the edge of the granuloma (Figure 6F). Conversely, the focal lesions of cattle that were not vaccinated were characterized by a diffuse membranous CD204+ immunolabeling (Figure 6F), with a complete absence of iNOS (Figure 6D).

The epithelioid cells forming multifocal lesions in both vaccinated and not vaccinated cattle showed positivity for TLR4 (Figure 1L and Figure 2O) and TLR9 (Figure 6B) in the cytoplasm, with membranous immunolabeling for CD204 (Figure 6F). Contrary to the uniform immunolabeling pattern observed for TLR9, the macrophages immunolabeled for TLR4 displayed a particular point-like labeling through the cytoplasm. Multifocal lesions were frequently surrounded by a high number of lymphocytes labeled for IFN-γ (Figure 1C and Figure 2C), with variable numbers of neutrophils displaying IFN-γ+ and iNOS+ immunolabeling in the cytoplasm present both inside and outside of the granulomas. Occasionally, the PMNs located inside the granulomas showed cytoplasmic immunolabeling for TLR2. Samples with diffuse paucibacillary forms showed similar features to the multifocal lesions (Figure 2D,L,P and Figure 6B,D,F), although TLR1 immunolabeling was occasionally observed in the cytoplasm of the epithelioid and Langhans giant cells forming the granulomas between the lymphocytic infiltrate (Figure 2H).

In the subcutaneous nodule formed at the site of the injection (Figure 3 and Figure 4), positive cells showed a similar distribution of the immunolabeling to those at the intestine. Few TLR1+ (Figure 3D–F), TLR2+ (Figure 3G–I) and TLR4+ (Figure 3J–L) macrophages with histiocytic appearance were intermingled between the granulomatous infiltrate. Epithelioid and Langhans giant cells forming the infiltrate showed positive immunolabeling for TLR9 (Figure 4A–C) and iNOS (Figure 4D–F) in the cytoplasm, with a more membranous labeling for CD204 (Figure 4G–I). Viable and degenerated iNOS+ neutrophils formed a necrotic core in some of the granulomatous lesions (Figure 4E). Variable numbers of positively immunolabeled lymphocytes for IFN-γ were present in close relationship to the epithelioid cells (Figure 3A–C).

### 3.2. Number of Immunolabeled Cells and Histological Scoring (H-Score) According to the Type of Lesion and Vaccination Status

According to our model selection procedure, for cell markers assessed in intestine sections, the most robust and parsimonious models—based on the fit quality indicators AIC (model with substantial support: ΔAIC < two units given the dataset) and W_i_—included an additive combination and interactions of all explanatory variables considered in the study (IS + SV + TL + IA + SI * SV + SV * TL + SV * IA). When these same markers were evaluated in the ISG and SLN, the best-classified models similarly included all the proposed explanatory variables (IS + SV + TL + IS * SV + TL * SV + TL * IS). The robustness of the results is underscored by multimodel inference [34]. The adjusted *R*^2^ estimates of the fitted models were high, ranging between 58.59% and 64.78%, indicating that the variables explained a large proportion of the variability observed in the different cellular subpopulations analyzed.

#### 3.2.1. Intestine

The total number of positively immunolabeled cells for TLR1, TLR2, TLR4 and IFN-γ detected at the intestine, though higher in the animals vaccinated with Silirum^®^ when compared with those not, did not show differences between the vaccination statuses (Appendix A).

When analyzing the interaction between infection status and vaccination status, no differences were observed in the counts of TLR1, TLR2, TLR4 and IFN-γ between vaccinated and unvaccinated cattle within each infection category (infected/non-infected) (the results of multiple comparisons can be seen in Appendix A). Within the control group, a significant increase (β = 0.752, SE = 0.111, *p* = 0.011) was observed in the number of macrophages immunolabeled against the TLR1 marker in the lamina propria of unvaccinated animals compared with vaccinated animals (Figure 5B). No differences were observed in the other markers analyzed in this same intestinal area or the other intestinal locations (Figure 5C–H). Finally, in the infected cows, vaccination with Silirum^®^ was associated with higher H-score values for iNOS (β_IS(IF)*VS(V)_ = 1.123, SE = 0.11, *p* < 0.001) and lower values for CD204 (β_IS(IF)*VS(V)_ = −0.421, SE = 0.291, *p* = 0.031) in the MAP-associated granulomas. The H-score TLR9 values, although slightly higher in vaccinated animals, were not significant (β_IS(IF)*VS(V)_ = 0.087, SE = 0.021, *p* = 0.991).

This interaction (TL * VS) was included in the best classified model, according to multi-model inference (Appendix A). Thus, to evaluate if the type of lesion present in the intestine significantly influenced the expression of the markers considered, samples with different pathological forms were compared between animals with the same vaccination status. Samples from cattle vaccinated with Silirum^®^ showed differences in the number of immunolabeled cells and H-score values according to the type of lesion present (control/no lesion, focal, multifocal). The number of positively immunolabeled macrophages for TLR1 was significantly higher in tissue sections presenting focal and multifocal lesions compared with those from the control animals (*p* < 0.001). The highest numbers of lymphocytes immunolabeled for IFN-γ (*p* < 0.001) and macrophages immunolabeled for TLR4 (*p* < 0.01) were detected in samples with multifocal lesions. Focal forms had significantly higher numbers of macrophages immunolabeled for TLR2 than those with multifocal lesions (*p* < 0.05), along with higher H-score values for iNOS and lower for CD204 within the granulomas (*p* < 0.001). For the TLR9 antibody, animals with multifocal lesions showed a significant increase (*p* < 0.001) in H-score values compared with animals with focal lesions. In non-vaccinated calves, a variation was also observed in the counts of the different markers analyzed depending on the lesion status. The highest number of positively immunolabeled cells for TLR1, TLR2 and TLR4 was detected in diffuse paucibacillary lesions (*p* < 0.001). Multifocal and diffuse forms showed significantly higher numbers of lymphocytes labeled for IFN-γ than the control and focal group (*p* < 0.001). Focal lesions showed greater immunolabeling for TLR4 and IFN-γ antibodies compared with the controls (*p* < 0.05), but lower than the multifocal and diffuse paucibacillary lesions (*p* < 0.001). The highest H-score values for TLR9 antibody were obtained in the multifocal lesions (*p* < 0.01), but no differences for iNOS or CD204 immunolabeling were detected between any of the lesions.

The effect of vaccination was evaluated among animal subsets (vaccinated/non-vaccinated) with the same type of lesion, considering the different intestinal areas (Figure 5 and Figure 6). Vaccination with Silirum^®^ was associated with a higher number of positively immunolabeled macrophages for TLR2 in samples with focal lesions, predominantly in the LP (β_VS(V)*IA(LP)_ = 0.321, SE = 0.047, *p* = 0.033) and the MLN (β_VS(V)*IA(MLN)_ = 0.139, SE = 0.050, *p* < 0.001). In these sections, a significant increase in lymphocytes labeled for IFN-γ was also detected in the LP of vaccinated animals (β_VS(V)*IA(LP)_ = 0.139, SE = 0.057, *p* = 0.039). The histological score for iNOS was higher in focal lesions from vaccinated calves (*p* < 0.01) compared with those from non-vaccinated ones (β_VS(V)_ = 0.439, SE = 0.088, *p* = 0.007), contrary to the H-score for CD204 (β_VS(V)_ = −0.183, SE = 0.037, *p* = 0.005). Immunized cattle with multifocal lesions exhibited a significantly higher number of TLR1-labeled macrophages compared with non-vaccinated calves (*p* < 0.05). Additionally, higher immunolabeling for TLR4 antibody was observed in the LP (β_VS(V)*IA(LP)_ = 0.282, SE = 0.015, *p* = 0.043) and GALT (β_VS(V)*IA(GALT)_ = 0.191, SE = 0.017, *p* = 0.034) of vaccinated cattle compared with non-vaccinated calves. However, no significant differences were detected in the number of immunolabeled cells or H-score values for TLR2, IFN-γ, TLR9, iNOS, or CD204 according to the vaccination status in animals with multifocal lesions.

#### 3.2.2. Scapular Lymph Node

The effect of vaccination influenced the observed variability of the different markers analyzed within the SLN. As the H-score values for TLR9, iNOS, and CD204 were only assessed in granulomatous lesions, and these were not present in the SLN nodes, these markers were not included as response variables in this location. A significant increase in TLR2 (*p* < 0.01) immunolabeling was observed at the SLN of the vaccinated animals while TLR4 (*p* < 0.05) and IFN-γ (*p* < 0.001) immunolabeling showed the opposite tendency. No differences in the number of immunolabeled cells for TLR1 antibody (*p* = 0.192) were observed between both vaccinate status (Appendix A).

Within the uninfected animals, the unvaccinated bovines presented a greater number of lymphocytes labeled for IFN-γ in the SLN compared with the vaccinated ones (*p* < 0.05). This trend was similar among infected animals that underwent vaccination. For the rest of the cellular markers analyzed, no significant differences were observed in their expression between vaccinated and non-vaccinated animals within the control and infected group (Appendix A).

As observed in the intestine, the interaction between vaccination status and type of lesion was included in the best classified model. Within the vaccinated animals, the number of cells labeled for IFN-γ in the SLN was significantly higher in animals with intestinal multifocal (*p* = 0.021) and focal lesions (*p* = 0.013) compared with the control group, with no differences observed between the focal and multifocal categories (*p* = 0.098). In turn, calves with multifocal forms showed greater numbers of macrophages labeled for TLR1 (*p* = 0.021) and TLR2 (*p* = 0.011) at the SLN than those with intestinal focal lesions (*p* < 0.05) or the controls (*p* < 0.01), which showed no differences between each other. Neither difference was observed for the other markers. Unvaccinated animals with focal lesions at the intestine presented a greater number of cells labeled for IFN-γ than uninfected animals (*p* = 0.001), with multifocal forms (*p* < 0.001) and diffuse paucibacillary lesions (*p* = 0.001). The latter also showed a greater number of interferons than the multifocal forms (*p* = 0.013). As for the rest of the markers, calves with diffuse paucibacillary lesions displayed higher numbers of macrophages labeled for TLR1 (*p* < 0.001) and TLR4 (*p* < 0.01) at the SLN than those with other types of lesions. Finally, the highest number of TLR2-positive cells in the SLN was observed in individuals with diffuse paucibacillary lesions at the intestine compared with the control group (*p* < 0.001). All results of multiple comparisons can be seen in Appendix A.

The interaction between vaccination status (vaccinated vs. unvaccinated) and the type of lesion present in the intestine only showed differences for the IFN-γ marker within the same lesion types. In this regard, unvaccinated cattle with focal (β_VS(NV)*TL(F)_ = 0.539, SE = 0.077, *p* < 0.001) and multifocal (β_VS(NV)*TL(MF)_ = 0.289, SE = 0.021, *p* = 0.012) lesions exhibited a higher number of IFN-γ labeled lymphocytes in the lymph node compared with vaccinated animals with the same type of lesion. No significant variations were observed in the expression of the rest of the biological markers in the SLN in relation to the combination of these variables.

#### 3.2.3. Injection Site—Granuloma

Calves infected and vaccinated with Silirum^®^ that had intestinal focal lesions showed a significantly higher number of macrophages marked for TLR2 (β_VS(V)*TL(F)_ = 0.239, SE = 0.031, *p* = 0.023) and lymphocytes marked for IFN-γ (β_VS(V)*TL(F)_ = 0.119, SE = 0.022, *p* = 0.043) in the ISG compared with those with multifocal forms (Figure 7A). Regarding the CD204 marker, the higher H-score values in the ISG in cattle with focal lesions was only significant compared with the group without lesions (β_VS(V)*TL(F)_ = 0.149, SE = 0.011, *p* = 0.035). An opposite trend was observed for the H-score values of TLR9 (β_VS(V)*TL(MF)_ = 0.349, SE = 0.041, *p* = 0.036) and iNOS (β_VS(V)*TL(MF)_ = 0.229, SE = 0.077, *p* = 0.001). In depth, animals with multifocal lesions exhibited a higher H-score for TLR9 at the ISG compared with the controls and those with focal lesions, and higher H-score for iNOS when compared with the control cows (Figure 7B). Finally, no additional differences were observed for TLR1 and TLR4 antibodies at this location, regardless of the type of intestinal lesion present (Figure 7A).

## 4. Discussion

The use of inactivated vaccines represents an effective way of reducing the incidence of clinical paratuberculosis [3,4,5], although it does not completely prevent the infection [6]. While serological responses of cattle vaccinated against MAP has been thoroughly studied [36,37,38,39], less attention has been given to the immunological events at the infection site. Recent evidence indicates that vaccination can modulate the host intestinal immune response [40]. In this present study, the histopathological examination of the intestinal samples taken at 330 DPV from calves vaccinated with Silirum^®^ and orally challenged with MAP revealed that most vaccinated animals (66.7%) had well-demarcated small granulomas restricted to the Peyer’s patches and regional lymph nodes, consistent with focal lesions, suggesting resilience to paratuberculosis [29,41]. In contrast, 50% of non-vaccinated and infected calves with MAP showed multifocal lesions, associated with progression of the infection [29,41], or even diffuse enteritis (16.7%), indicating the onset of early clinical disease.

TLR1, TLR2, and TLR4 are expressed on the cell surface [11]. Nonetheless, positive cells consistently showed cytoplasmic immunolabeling, as observed in previous immunohistochemical studies [13,42,43,44]. This may reflect a redistribution of membrane proteins into the cytoplasm due to antigen retrieval [45] or antibodies binding to TLR proteins in the process of synthesis or trafficking within the endoplasmic reticulum and Golgi apparatus [46]. Alternatively, TLR4 is known to undergo internalization into endosomes following activation [47]. The majority of macrophages forming multifocal and diffuse lesions exhibited a distinctive dot-like cytoplasmic immunolabeling pattern for TLR4. These cells did not show evidence of pigment accumulation (e.g., hemosiderin or lipofuscin) on hematoxylin-eosin-stained sections, and no similar labeling was seen in negative controls. Therefore, this could be indicative of TLR4-containing endosomes associated with TLR4 activation in these lesions, as proposed in a previous immunohistochemical analysis of intestinal lesions from naturally infected cattle [13].

Vaccinated calves controlling the infection inside focal lesions displayed increased numbers of intestinal macrophages immunolabeling TLR2 and M1 polarization of the granulomas. Despite representing a well-recognized target of MAP mechanisms of immune subversion in in vitro conditions [48], TLR2 is considered the most important TLR involved in mycobacteria recognition and a general marker of the M1 phenotype [22,49,50]. Therefore, the upregulation of this PRR by the macrophages located outside of the granulomas suggests a robust activation of these cells, which may support the contention of MAP inside the focal lesions [51]. The scant bacilli present in this type of lesion are typically located inside the macrophages [29], thereby explaining the immunolabeling observed for intracellular TLR9 [52], but not for surface TLR1, TLR2 or TLR4. TLR9 signaling promotes the secretion of IL-12 by murine macrophages in response to in vitro infection with *Mycobacterium tuberculosis* and contributes to an adaptive Th1 immunity [14]. Therefore, the lysis of MAP inside the macrophages and the activation of TLR9 by the DNA released could trigger IL-12 production in the granulomas, with subsequent differentiation of nearby Th1 lymphocytes immunolabeling IFN-γ [14,52,53].

The presence of IFN-γ-labeled lymphocytes around the granulomas aligns with previous immunohistochemical studies and suggests a role in activating lesion-forming macrophages [21]. Both pro- and anti-inflammatory macrophage phenotypes coexisted inside the focal lesions of the vaccinated cattle, with M2-polarized (CD204+) macrophages localized at the periphery, and M1-polarized (iNOS+) cells in the center. This spatial distribution of macrophages within the granuloma has been previously reported for latent *M. tuberculosis* infection in humans [54], and is probably key for a successful control of the pathogen with limited damage to the host tissues [54,55,56]. According to these results, most of the vaccinated calves mount a coordinated innate and cell-mediated immune response in the intestine capable of controlling MAP infection through the formation of focal lesions restricted to the Peyer’s patches and the mesenteric LN.

Calves vaccinated with Silirum^®^ showing multifocal forms displayed reduced intestinal immunolabeling for TLR2, high IFN-γ immunolabeling and diffuse CD204+ labeling of the lesions, like their unvaccinated counterparts. Here, the number of immunolabeled cells for IFN-γ was directly associated with the severity of the granulomatous lesions rather than with the vaccination status. Previous studies have identified upregulated IFN-γ gene expression in the intestine of naturally infected cows with clinical paratuberculosis [8,10,19]. These findings indicate that IFN-γ does not always play a protective role during MAP infection, so its interpretation must be contextualized based on the type of lesion and the immune microenvironment. Considering that MAP can inhibit IFN-γ-mediated activation of bovine monocytes infected in vitro [57], a similar mechanism could operate within the multifocal and diffuse lesions. Early deficient innate response may facilitate MAP establishment in the macrophages and persistent infection [58], evidenced by M2-polarized granulomatous lesions and an ineffective IFN-γ response that potentially leads to CD4+ T cell exhaustion as shown by immunohistochemistry and flow cytometry analyses of intestinal tissues from naturally infected cattle with clinical signs [23,24,59]. The observation of neutrophils immunolabeling IFN-γ in samples with multifocal and, to a greater extent, diffuse lesions could suggest a potential role of these cells in the production of IFN-γ or phagocytosis during the late phases of the disease [60,61], when the population of CD4+ T cell starts to decline [59]. Further, the neutrophils present inside these lesions also expressed iNOS, arguably as a compensatory mechanism to overcome the lack of activation of the macrophages [62].

Beside TLR9 and CD204 immunolabeling, epithelioid macrophages forming the multifocal and diffuse granulomatous lesions displayed a particular point-like labeling through the cytoplasm for TLR4, regardless of the vaccination status, which colocalized with MAP bacilli immunolabeling using a customized antibody. TLR4 participate in *Mycobacterium tuberculosis* phagocytosis by macrophages in vitro [63], so it could likewise participate in the phagocytosis of extracellular bacteria released during the lysis of heavily infected macrophages in MAP-associated granulomas, as proposed in a recent immunohistochemical analysis of paratuberculosis lesions from naturally infected cattle [13]. Nevertheless, the endocytosis of TLR4 leads to signaling through a TRIF-adaptor protein and the production of type I IFNs and IL-10 [47,64], which favor an anti-inflammatory micro-environment suitable for the proliferation of MAP [65,66]. Further, the epithelioid cells forming the granulomatous lesions in sections with diffuse lymphocytic forms occasionally expressed TLR1, supporting the presence of extracellular bacteria or bacterial components [13,51]. According to these results, most of the unvaccinated calves and some of the vaccinated calves developed an ineffective anti-inflammatory response that was unable to restrict MAP growth, leading to a widespread growth of the granulomas, in turn resulting in multifocal and diffuse lesions.

Non-vaccinated calves with focal lesions (33.3%) showed low levels of TLR2 immunolabeling at the intestine, along with a diffuse CD204 labeling of the granulomas, similar to those with progressive multifocal forms. Thus, these may in fact represent animals with progression of MAP infection, but at an earlier stage of the disease. This hypothesis would justify the still moderate production of IFN-γ due to the yet mild severity of the lesions.

The refined oil-based adjuvant of Silirum^®^ allows a homogenous distribution of the vaccine at the site of the injection and an efficient antigen presentation in vaccinated lambs, evidenced by the presence of small adjuvant droplets and strong MHC-II immunolabeling [67]. In our study, variations in TLR2 and IFN-γ immunolabeling were observed in the ISG according to the type of intestinal lesion, although no such association could be made for the draining of the SLN. Specifically, vaccinated calves with focal lesions displayed higher numbers of histiocytes immunolabeling TLR2 and lymphocytes labeled for IFN-γ at the ISG compared with those with multifocal forms. These findings suggest a persistent antigenic stimulation and the development of a local Th1 immune response at the site of the injection, which appear stronger in animals capable of controlling the intestinal infection. Therefore, the efficacy of Silirum^®^ is probably linked to the capacity of host’s innate immune cells to respond to vaccine antigens, which may be associated with individual variations in specific TLR genes [12].

The present study provides an insight into the mucosal immune response of calves vaccinated with Silirum^®^ and experimentally infected with paratuberculosis. The presence of focal lesions in the Peyer’s patches and lymph nodes of most vaccinated calves could be the reflection of an effective proinflammatory response at the site of the infection, where the animal can control the growth of MAP inside M1-polarized granulomas. TLR9 activation may trigger IL-12 secretion by lesion-forming macrophages, leading to IFN-γ production by nearby lymphocytes. TLR2 upregulation by intestinal macrophages suggest an activation state of these cells that could prevent lesion progression. On the other hand, the presence of multifocal lesions in most of the unvaccinated calves and a small percentage of those vaccinated probably reflect an ineffective anti-inflammatory response, with reduced macrophage activation and M2 polarization of the granulomas, despite an intense production of IFN-γ. TLR4 signaling in these lesions may contribute to further anti-inflammatory cytokine production. Differences in the outcome of MAP infection among vaccinated calves suggest individual factors affecting the host response to Silirum^®^, evidenced by variations in the intensity of the inflammatory response developed at the injection site.

## 5. Conclusions

Herein we provide evidence that the protection conferred by Silirum^®^ vaccine in calves experimentally infected with paratuberculosis is associated with an enhanced pro-inflammatory response in the intestine. The crosstalk between the innate and adaptive immunity seems key for the development of a cell-mediated immune response capable of controlling MAP infection within the granulomas. Nonetheless, IFN-γ does not always act as a protective marker in the context of vaccination and its interpretation must be contextualized based on the polarization status of the macrophages. The progression of the disease in some vaccinated calves indicates that there are individual factors that modulate the host response to Silirum^®^ and influence the outcome of the infection. In this sense, the analysis of the inflammatory response developed at the site of the injection could represent a useful tool to assess the efficacy of vaccination.

Further investigation is required to identify the precise molecular and cellular processes implicated in the protection conferred by Silirum^®^.

## Figures and Tables

**Figure 1 animals-15-01841-f001:**
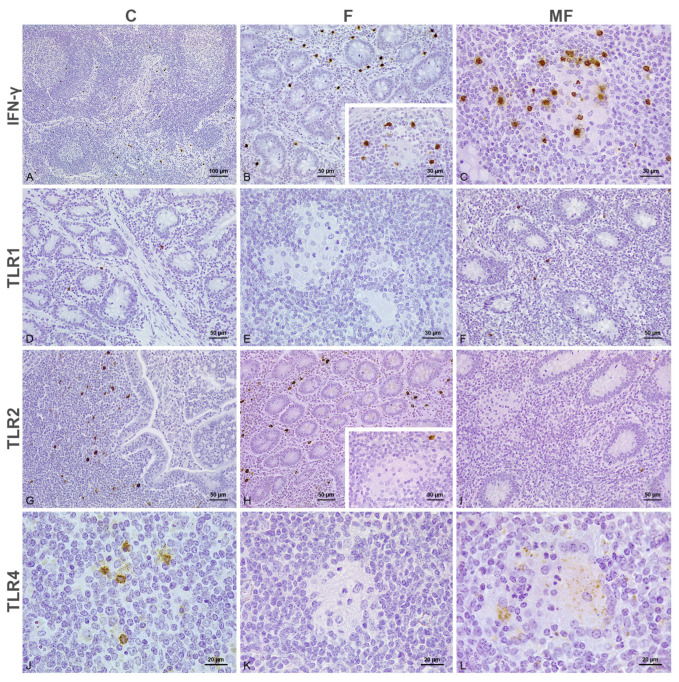
Tissue sections of the small intestine and associated lymph nodes of calves vaccinated with Silirum^®^ immunolabeled for IFN-γ, TLR1, TLR2 and TLR4 antibodies. **Panels** (**A**–**C**). IFN-γ immunohistochemistry. (**A**) Positively immunolabeled lymphocytes for IFN-γ concentrating in the cortico-medullary junction of a mesenteric lymph node from an uninfected control. (**B**) Moderate numbers of lymphocytes immunolabeled for IFN-γ in the jejunal lamina propria of a calf with focal lesions. Inset: few labeled lymphocytes surrounding a granuloma present in the Peyer’s patches from the same animal. (**C**) Intense IFN-γ immunolabeling in close relationship to a multifocal lesion in the ileum. **Panels** (**D**–**F**). TLR1 immunohistochemistry. (**D**) Scant positively immunolabeled macrophages with histiocytic appearance for TLR1 scattered between the crypts of Lieberkühn in the intestinal mucosa from a control animal. (**E**) Lack of TLR1 immunolabeling in the focal lesions situated in the jejunal Peyer’s patches. (**F**) Absence of TLR1 immunolabeling in the epithelioid cells forming a multifocal lesion in the jejunal LP, with scant number of TLR1+ macrophages nearby. **Panels** (**G**–**I**). TLR2 immunohistochemistry. (**G**) Positively immunolabeled macrophages for TLR2 concentrate in the dome of a Peyer’s patch from a control calf. (**H**) Moderate numbers of histiocytes labeled for TLR2 in the LP of an animal with focal lesions in the GALT. Inset: absence of TLR2 immunolabeling in a granuloma from the same animal, with a positive macrophage nearby. (**I**) Lack of TLR2 immunolabeling in a multifocal lesion situated in the LP, with scarce TLR2+ macrophages in the surrounding tissue. **Panels** (**J**–**L**). TLR4 immunohistochemistry. (**J**) Positively immunolabeled macrophages for TLR4 in the interfollicular region of a Peyer’s patch from an uninfected calf. (**K**) Lack of TLR4+ immunolabeling in a focal lesion located in the paracortex of a mesenteric lymph node. (**L**) Distinctive point-like immunolabeling for TLR4 in the cytoplasm of the epithelioid and Langhans giant cells forming a multifocal lesion. C: control; F: focal; MF: multifocal.

**Figure 2 animals-15-01841-f002:**
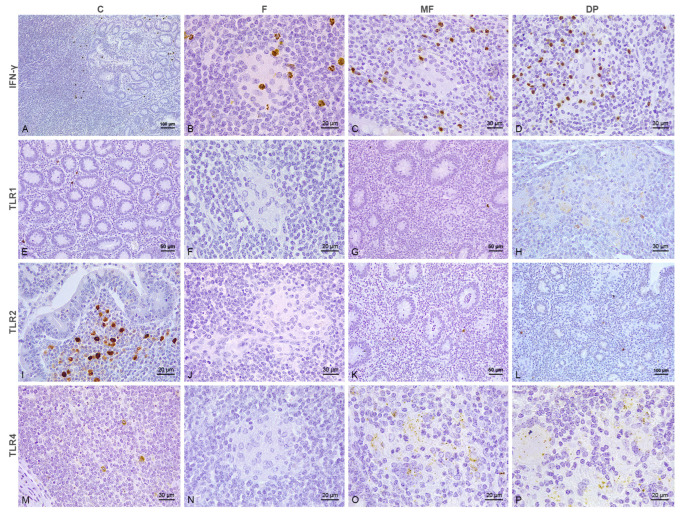
Tissue sections of the intestine and regional lymph nodes from unvaccinated calves immunolabeled for IFN-γ, TLR1, TLR2 and TLR4 antibodies. **Panels** (**A**–**D**). IFN-γ immunohistochemistry. (**A**) Positively immunolabeled lymphocytes for IFN-γ concentrated in the dome of the Peyer’s patches of control cattle. (**B**) Mild numbers of lymphocytes labeled for IFN-γ in close relationship with a focal lesion situated in the jejunal Peyer’s patches. (**C**) Moderate numbers of lymphocytes immunolabeled for IFN-γ surrounding a multifocal lesion situated in the lamina propria. (**D**) Intense IFN-γ immunolabeling associated with the presence of a granulomatous lesion from an animal with diffuse lymphocytic forms. Few immunolabeled PMNs consistent with neutrophils are present inside the lesion. **Panels** (**E**–**H**). TLR1 immunohistochemistry. (**E**) Few positively immunolabeled histiocytic macrophages scattered between the crypts in the lamina propria of an uninfected calf. (**F**) No TLR1+ macrophages in a focal lesion situated in a mesenteric lymph node. (**G**) Absence of TLR1 immunolabeling in the multifocal lesions and scarce immunolabeling of the surrounding macrophages. (**H**) Positive immunolabeling for TLR1 in the cytoplasm of the epithelioid and Langhans giant cells forming a granulomatous lesion in an animal with diffuse enteritis. **Panels** (**I**–**L**). TLR2 immunohistochemistry. (**I**) High numbers of immunolabeled macrophages for TLR2 clustering at the dome of a Peyer’s patch from a control animal. (**J**) Absence of TLR2 immunolabeling in a focal lesion located in a mesenteric lymph node. (**K**) Lack of TLR2 immunolabeling in the macrophages forming the multifocal lesions, with scarce immunolabeling of those situated outside of the granulomas. (**L**) Few TLR2+ histiocytic macrophages appear intermingled between the granulomatous lesions of a calf with diffuse lymphocytic forms, which do not immunolabel for TLR2. **Panels** (**M**–**P**). TLR4 immunohistochemistry. (**M**) Few positively immunolabeled macrophages situated in the cortex of a mesenteric lymph node. (**N**) Lack of immunolabeling for TLR4 in a focal lesion located in the jejunal Peyer’s patches. (**O**) Granular dot-like immunolabeling for TLR4 in the cytoplasm of the macrophages from a multifocal granuloma. (**P**) Similar labeling pattern to that described in the previous image in the epithelioid and Langhans giant cells forming a diffuse granulomatous lesion. C: control; F: focal; MF: multifocal; DP: diffuse paucibacillary.

**Figure 3 animals-15-01841-f003:**
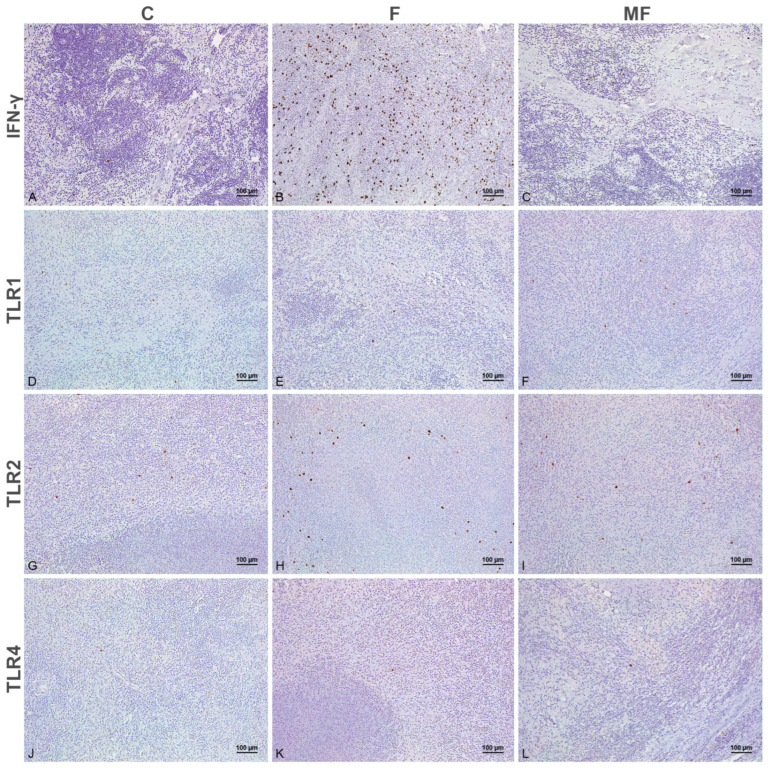
Tissue sections of the subcutaneous granuloma formed at the site of the injection immunolabeled for IFN-γ, TLR1, TLR2 and TLR4, 100× magnification. **Panels** (**A**–**C**). IFN-γ immunohistochemistry. Higher numbers of positively immunolabeled lymphocytes for IFN-γ among the granulomatous infiltrate in calves with intestinal focal lesions (**B**) compared with those uninfected (**A**) or with multifocal forms (**C**). **Panels** (**D**–**F**). TLR1 immunohistochemistry. Similar immunopositivity for TLR1 antibody in the histiocytic macrophages present between the epithelioid and Langhans cells in control (**D**) and infected cattle with focal (**E**) and multifocal (**F**) lesions. **Panels** (**G**–**I**). TLR2 immunohistochemistry. Greater numbers of histiocytes labeled for TLR2 in calves with focal lesions in the Peyer’s patches (**H**), compared with the control (**G**) and multifocal (**I**) groups. **Panels** (**J**–**L**). TLR4 immunohistochemistry. Rare TLR4+ histiocytes are present among the granulomatous infiltrate in control (**J**) and infected cattle with either focal (**K**) and multifocal (**L**) lesions. C: control; F: focal; MF: multifocal.

**Figure 4 animals-15-01841-f004:**
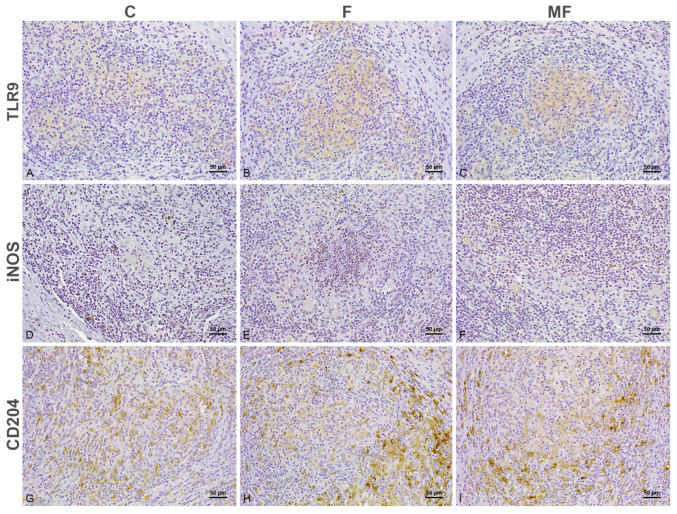
Tissue sections of the subcutaneous nodule formed after vaccination immunolabeled for TLR9, iNOS and CD204, 200x magnification. **Panels** (**A**–**C**). TLR9 immunohistochemistry. Moderate immunolabeling in the granulomatous infiltrate forming the injection-site granuloma of control (**A**) and infected cattle with focal (**B**) and multifocal lesions (**C**). **Panels** (**D**–**F**). iNOS immunohistochemistry. Mild to moderate immunolabeling in the cytoplasm of the epithelioid and Langhans giant cells infiltrating the subcutaneous tissue in all three groups (**D**–**F**), with fewer immunopositivity in control cattle (**D**). **Panels** (**G**–**I**). CD204 immunohistochemistry. Epithelioid macrophages forming the post-vaccination granuloma in calves with focal lesions (**G**) show more intense labeling for CD204, especially in the margins of the granulomas, compared with control (**H**) and infected cattle with multifocal lesions (**I**). C: control; F: focal; MF: multifocal.

**Figure 5 animals-15-01841-f005:**
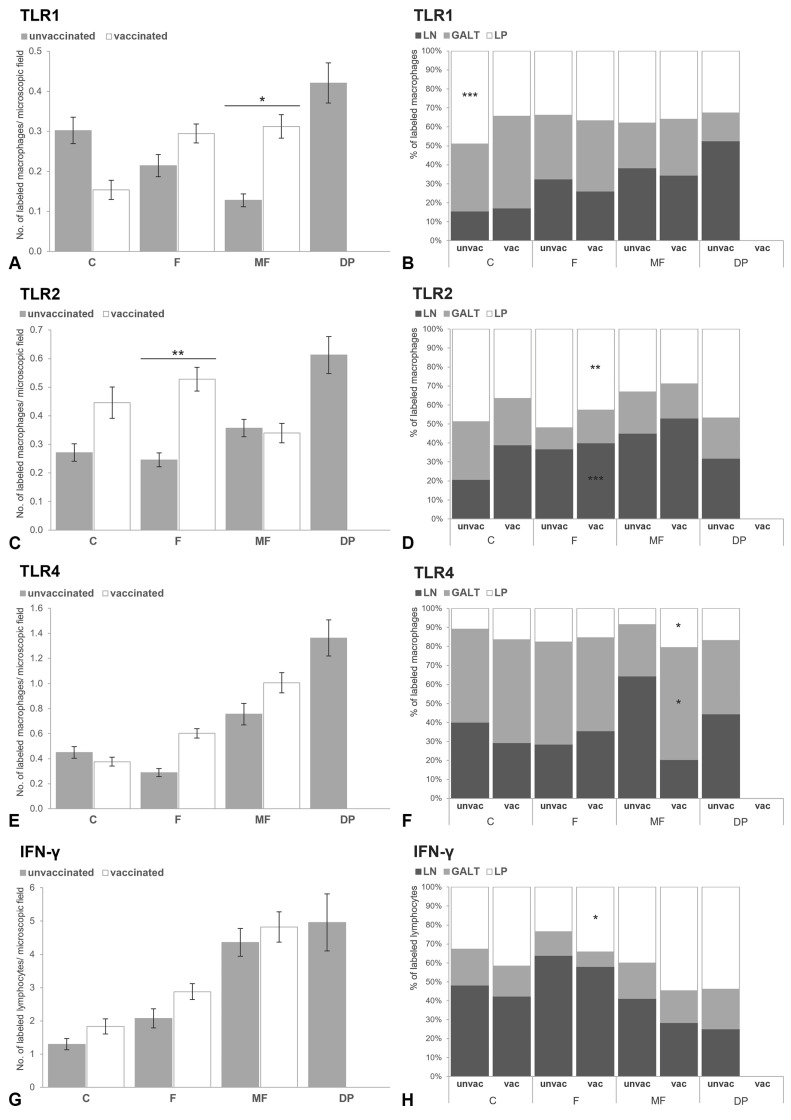
Bar charts indicating the mean number of immunolabeled cells at the intestine for TLR1 (**A**), TLR2 (**C**), TLR4 (**E**) and IFN-γ (**G**) according to the type of lesion present (F: focal; MF: multifocal; DP: diffuse paucibacillary) and the immunization status (unvaccinated; vaccinated). Superscript asterisks indicate statistical significance between vaccinated and not vaccinated cattle with the same type of lesion (* *p* < 0.05; ** *p* < 0.01; *** *p* < 0.001). Error bars: standard error. 100% stacked bar charts showing the percentage of the immunolabeled cells for TLR1 (**B**), TLR2 (**D**), TLR4 (**F**) and IFN-γ (**H**) in the different intestinal locations (LP, lamina propria; GALT, gut-associated lymphoid tissue; LN, lymph node) based on the lesion category and the status of vaccination. Asterisks inside the box indicate statistical significance in the corresponding location between vaccinated and not vaccinated calves in the same lesion category (* *p* < 0.05; ** *p* < 0.01; *** *p* < 0.001).

**Figure 6 animals-15-01841-f006:**
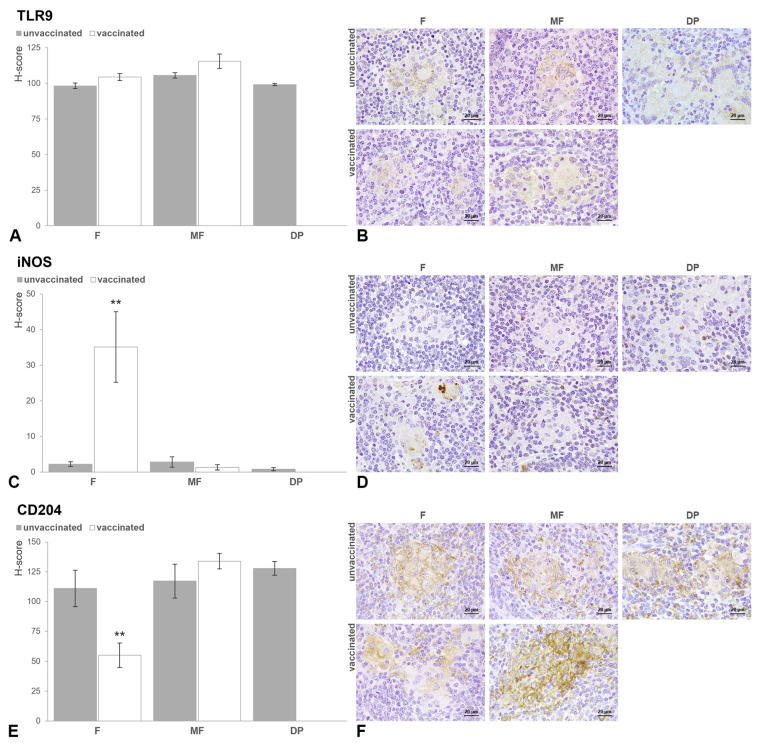
Bar charts showing the mean H-score values for TLR9 (**A**), iNOS (**C**) and CD204 (**E**) antibodies for each lesion category (F, focal; MF, multifocal; DP, diffuse paucibacillary) according to the vaccination status (unvaccinated; vaccinated). Superscript asterisks indicate statistical significance (** *p* < 0.01). Error bars: standard error. Microphotographs of TLR9 (**B**), iNOS (**D**) and CD204 (**F**) immunolabeling in each type of lesion (F, focal; MF, multifocal; DP, diffuse paucibacillary) based on the immunization status of the animal (unvaccinated; vaccinated). (**B**) Uniform cytoplasmic immunolabeling for TLR9 in all of the granulomatous lesions analyzed. (**D**) Epithelioid and Langhans giant cells forming focal lesions in cattle vaccinated with Silirum^®^ show granular immunolabeling for iNOS in the cytoplasm, against the other groups. Few positively immunolabeled neutrophils are present both inside and outside of the multifocal and diffuse lesions. (**F**) Moderate membranous and cytoplasmic CD204+ immunolabeling of the macrophages situated in the margins of the focal lesions from vaccinated calves, compared with the diffuse immunolabeling of the granuloma in the rest of the groups.

**Figure 7 animals-15-01841-f007:**
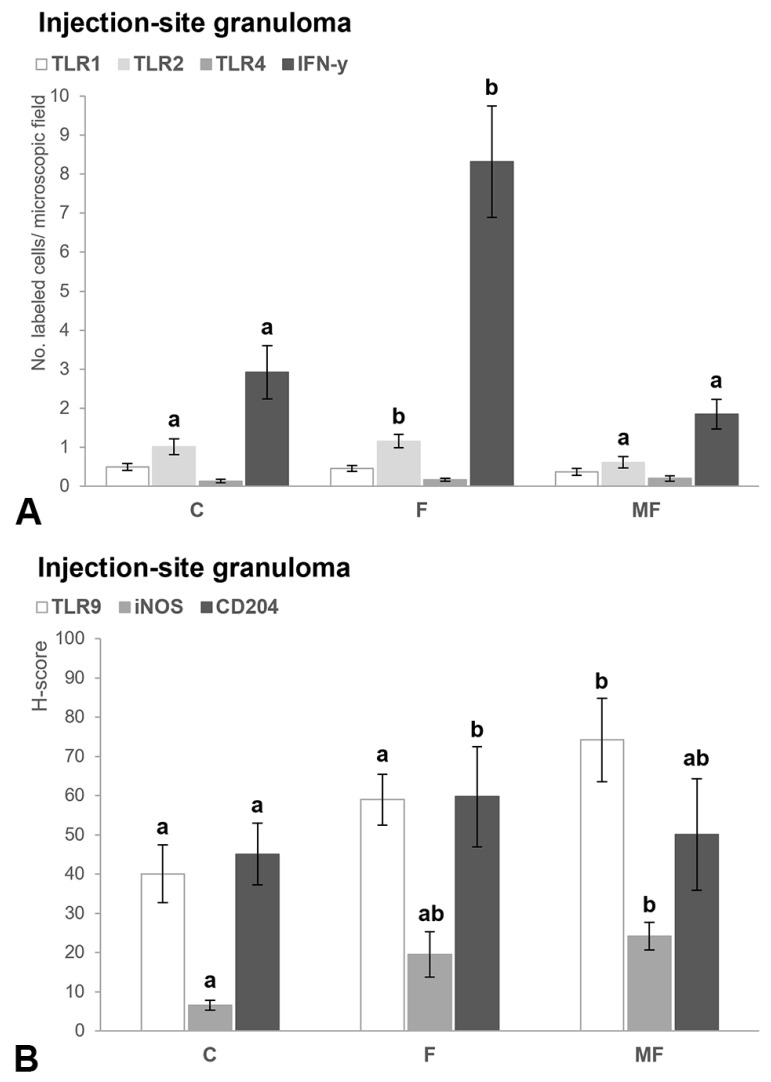
(**A**) Bar chart showing the mean number of immunolabeled macrophages for TLR1, TLR2, TLR4 and lymphocytes for IFN-γ at the post-vaccination subcutaneous nodule according to the type of lesion present at the intestine (C, control, F, focal; MF, multifocal). Superscript letters indicate statistical significance. Error bars: standard error. (**B**) Bar chart indicating the mean histological score (H-score) for TLR9, iNOS and CD204 in the granuloma formed at the site of the injection based on the lesion category. Superscript letters indicate statistical significance. Error bars: standard error.

## Data Availability

The original contributions presented in this study are included in the article and Appendix A. Further inquiries can be directed to the corresponding authors.

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
