# Peer review of "Influence of Paratuberculosis Vaccination on the Local Immune Response in Experimentally Infected Calves: An Immunohistochemical Analysis"

_animals, 2025, doi:10.3390/ani15131841_

Round 1
Reviewer 1 Report
Comments and Suggestions for Authors
This study investigated in detail the local immune response at the intestinal level in calves vaccinated with Silirum®, an inactivated vaccine used against bovine paratuberculosis, and subsequently experimentally infected with MAP. It addresses an important gap: the immune response at the site of infection (the gut) is much less studied than the systemic response.
The abstract clearly outlines the aim of the study, specifying both the techniques studied and the tissues analysed, and the classification of lesions. With respect to the statement “Silirum®-induced protection is associated with an enhanced intestinal immune response” should be better specified with respect to the results collected. The authors are also asked to refer to the limitations of vaccines with respect to the fact that animals with multifocal lesions, irrespective of vaccination, have a similar immune profile, can be seen as a limitation of the vaccine.
INTRODUCTION:
The introduction in general appears well written and structured, but can be improved. In the section discussing immune responses, it might be interesting to elaborate on why it is important to study them. In this regard, the authors are advised to focus on the intestinal site as the true field of paratuberculosis.
It would be useful to explain in one sentence that TLRs are more related to activation (early recognition), whereas iNOS/CD204 reflect
the actual function of macrophages (M1/M2 respectively). Also add references to this.
The materials and methods and results section are well described, the figures are very useful.
The discussion section is well structured in fact it manages to describe the results well. You cover many aspects of the immune response: TLRs, IFN-γ, iNOS, CD204, macrophage polarisation, presence of neutrophils, interpretation of lesions.
Link the histological data well with the potential immunological mechanisms underlying the lesions.
In order to make the discussion clearer, it is suggested that you state the main findings of the study now so that the discussion can be well structured.
You can better describe the IFN-γ part, in fact this is presented as an indicator of a good response, other times as a marker of an ineffective or exhausted immune effort.
A final paragraph integrating the results into a coherent model of immune response to Silirum® vaccine is suggested.
Author Response
First of all, we would like to thank the editor and reviewers for their additional comments and suggestions that have contributed to improve even more the quality of the first revision. We have carefully revised the manuscript and addressed all the comments and corrections suggested.
- The introduction in general appears well written and structured, but can be improved. In the section discussing immune responses, it might be interesting to elaborate on why it is important to study them. In this regard, the authors are advised to focus on the intestinal site as the true field of paratuberculosis.
Author´s response:
We appreciate the reviewer’s insightful comment. In response, we have revised the Introduction to emphasize the importance of investigating immune responses specifically at the intestinal level, which represents the primary site of Mycobacterium avium subsp. paratuberculosis (MAP) infection. A statement highlighting the relevance of this localized immune response in understanding the pathogenesis and progression of paratuberculosis has been added accordingly.
- It would be useful to explain in one sentence that TLRs are more related to activation (early recognition), whereas iNOS/CD204 reflect the actual function of macrophages (M1/M2 respectively). Also add references to this.
Author´s response:
We thank the reviewer for this valuable suggestion. In response, we have added two concise sentences to clarify the distinct roles of these markers: Toll-like receptors (TLRs) are primarily involved in early pathogen recognition and activation of innate immunity, while iNOS and CD204 are indicative of macrophage polarization towards M1 and M2 functional phenotypes, respectively. Relevant references have also been included to support this distinction.
- The materials and methods and results section are well described, the figures are very useful.
Author´s response:
We acknowledge the reviewer’s comment and we sincerely appreciate your recognition of the efforts invested in this part of the manuscript.
- The discussion section is well structured in fact it manages to describe the results well. You cover many aspects of the immune response: TLRs, IFN-γ, iNOS, CD204, macrophage polarisation, presence of neutrophils, interpretation of lesions.
Author´s response:
We greatly appreciate your favorable evaluation of the discussion section.
- Link the histological data well with the potential immunological mechanisms underlying the lesions.
Author´s responses
We appreciate the reviewer’s observation. In response, we have revised the Discussion to more clearly integrate the histological findings with the underlying immunological mechanisms. Specifically, we have added statements throughout the section that link the observed tissue alterations with the local immune responses, aiming to provide a more cohesive interpretation of how immunological dynamics may contribute to lesion development and progression.
- In order to make the discussion clearer, it is suggested that you state the main findings of the study now so that the discussion can be well structured.
Author´s response:
We acknowledge the reviewer’s comment. At the beginning of each paragraph of the discussion, the main results are cited to better contextualize the subsequent discussion. At the end of the discussion a final paragraph was added integrating the main findings of the study in a model of the immune response that takes in the intestine of these animals.
- You can better describe the IFN-γ part, in fact this is presented as an indicator of a good response, other times as a marker of an ineffective or exhausted immune effort.
Author´s response:
We thank the reviewer for this important observation. To address it, we have expanded the discussion on IFN-γ to reflect its context-dependent role in the immune response. Specifically, we now clarify that while IFN-γ is generally associated with a protective Th1-type response, its sustained expression in chronic stages of infection may also indicate immune dysregulation or exhaustion. This nuanced interpretation has been incorporated into the revised manuscript to provide a more accurate and balanced view.
- A final paragraph integrating the results into a coherent model of immune response to Silirum® vaccine is suggested.
Author´s response:
We appreciate the reviewer’s constructive suggestion. In response, we have added a concluding paragraph that presents an integrated model of the immune response elicited by Silirum® vaccination. This synthesis brings together our key findings—to propose a coherent framework that may help explain the vaccine’s immunological effects and its potential role in disease modulation.
Reviewer 2 Report
Comments and Suggestions for Authors
•In the Discussion, some paragraphs are dense. Breaking them into shorter sections with subheadings (e.g., “TLR2-related responses”, “Macrophage polarization”) could help readability.
Figures 1 and 2 do not mention the magnification. Sections shown are at different magnification which makes it difficult to compare the IHC expression or different biomarkers in different treatment groups, please consider providing all photomicrographs at same magnification
Comments on the Quality of English LanguageThere are some awkward phrasings and minor grammatical issues that could benefit from language polishing. Examples:
•Line 27: “Vaccination remains as the most cost-effective…” → “Vaccination remains the most cost-effective…”
•Line 75: “The aim of this study was to evaluate…” → Consider rephrasing to “This study aimed to evaluate…” for conciseness.
Some results are repeated across text and figures, which could be streamlined for better flow.
Author Response
First of all, we would like to thank the editor and reviewers for their additional comments and suggestions that have contributed to improve even more the quality of the first revision. We have carefully revised the manuscript and addressed all the comments and corrections suggested.
- In the Discussion, some paragraphs are dense. Breaking them into shorter sections with subheadings (e.g., “TLR2-related responses”, “Macrophage polarization”) could help readability.
Author´s response:
We thank the reviewer for this helpful suggestion. In response, we have revised the Discussion section by subdividing dense paragraphs into shorter, more focused segments. Where appropriate, we have also introduced subheadings (e.g., “TLR2-related responses”, “Macrophage polarization”) to improve the structure and enhance overall readability for the reader.
- Figures 1 and 2 do not mention the magnification. Sections shown are at different magnification which makes it difficult to compare the IHC expression or different biomarkers in different treatment groups, please consider providing all photomicrographs at same magnification.
Author´s response:
We appreciate the reviewer’s observation regarding the magnification used in Figures 1 and 2. For the purposes of this study, we aimed to capture both the cellular-level detail of immunolabeling and the broader distribution of immunopositive cells within the tissue. To achieve this, we selected different magnifications depending on the specific feature we intended to highlight—for example, higher magnifications were used to better visualize intracellular staining patterns (e.g., Figure 1L, Figure 2O–P), while lower magnifications were used to show spatial distribution within lesions or across tissue sections (e.g., Figure 1A, Figure 2E, 2L). To support comparability despite these differences, scale bars have been included in all panels. These provide clear spatial reference points and allow the reader to interpret relative size and distribution accurately.
Quantality of the English
- There are some awkward phrasings and minor grammatical issues that could benefit from language polishing. Examples:
Line 27: “Vaccination remains as the most cost-effective…” → “Vaccination remains the most cost-effective…”
Line 75: “The aim of this study was to evaluate…” → Consider rephrasing to “This study aimed to evaluate…” for conciseness.
Author´s response:
We acknowledge the reviewer’s comments and these grammatical issues were amended in the text.
- Some results are repeated across text and figures, which could be streamlined for better flow.
Author´s response:
We thank the reviewer for this thoughtful observation. While we acknowledge that some overlap exists between the main text and figure legends, this was an intentional choice to ensure that each figure—particularly the immunohistochemical images—can be understood independently. In histological studies, it is standard practice to provide sufficient explanatory detail in figure legends to facilitate standalone interpretation. That said, we have carefully reviewed the manuscript to minimize unnecessary repetition and ensure that any duplication serves a clear purpose in aiding clarity and comprehension.
Reviewer 3 Report
Comments and Suggestions for Authors
The author found that vaccinated animals with disease progression and unvaccinated patients with multifocal or diffuse lesions exhibited intestinal weakness The anti-inflammatory response in Nete's immune system and granulomas. In short, vaccination Silirum ® Enhancing intestinal immune response may help protect. I think the research on anti paratuberculosis has a confusing logic and significant issues.
- The title of the article should be concise and not expressed clearly.
- The sample was taken from the granuloma at the injection site,such as lymph nodes, glomeruli, mesenteric lymph nodes, and intestines. The focal form of infection control in vaccinated calves exhibits strong innate reactions in the intestine, and is cytokine detection in granulomas specific and comprehensive?
- Why are the scalebars in the picture different in size?
- The conclusion really needs to be refined, it's too cumbersome and not precise enough.
- Do many of the calculation methods in the article have reference support?
- Most of the references are too old, and it is advisable to cite the latest literature from the past three years as much as possible.
- At the very least,authors should provide a detailed introduction to Silirum ®.
Author Response
First of all, we would like to thank the editor and reviewers for their additional comments and suggestions that have contributed to improve even more the quality of the first revision. We have carefully revised the manuscript and addressed all the comments and corrections suggested.
- The title of the article should be concise and not expressed clearly.
Author´s response:
We appreciate the reviewer’s suggestion regarding the clarity and conciseness of the title. In response, we have revised the title to more clearly and succinctly reflect the content and focus of the study. The updated title, “Influence of paratuberculosis vaccination on the local immune response in experimentally infected calves: An immunohistochemical analysis”, better communicates the scope and methodology of the work.
- The sample was taken from the granuloma at the injection site, such as lymph nodes, glomeruli, mesenteric lymph nodes, and intestines. The focal form of infection control in vaccinated calves exhibits strong innate reactions in the intestine, and is cytokine detection in granulomas specific and comprehensive?
Author´s response:
We thank the reviewer for this important question. Previous studies of natural paratuberculosis infection have shown that focal lesions are associated with the production of pro-inflammatory cytokines characteristic of the M1 macrophage phenotype, such as TNF-α (Fernández et al., 2017). In our study, IFN-γ was the only cytokine analyzed and was predominantly produced by lymphocytes rather than macrophages. Based on our immunolabeling patterns and supported by existing literature (Bafica et al., 2005; Chang et al., 2007), we hypothesize that macrophages within focal lesions may produce pro-inflammatory IL-12 via a TLR9-dependent pathway, whereas those in multifocal lesions could secrete anti-inflammatory IL-10 through TLR4 signaling. We acknowledge that a comprehensive analysis of cytokine profiles within granulomas would greatly enhance understanding of the immune response and consider this an important avenue for future research.
- Why are the scalebars in the picture different in size?
Author´s response:
We appreciate the reviewer’s attention to detail. In Figures 3, 4, and 6, the scale bars are consistent in length because the images within each figure were captured at the same magnification. In contrast, Figures 1 and 2 contain microphotographs taken at varying magnifications to highlight different features, which naturally results in scale bars representing different actual distances (100, 50, 30, or 20 μm). These distances were carefully chosen to maintain visual consistency, so the scale bars appear similar in size across panels despite reflecting different spatial scales. This approach allows accurate interpretation of image dimensions while preserving clarity
- The conclusion really needs to be refined, it's too cumbersome and not precise enough.
Author´s response:
We acknowledge the reviewer comment and the conclusion section was modified for better clarity.
- Do many of the calculation methods in the article have reference support?
Author´s response:
We appreciate the reviewer’s comment. We would like to clarify that all statistical analyses performed in this study are supported by well-established references (Burnham and Anderson, 2002; Whittingham et al., 2006; Zuur et al., 2007, 2010), which are duly cited in the Statistical Analysis section of the Methods. Nevertheless, if there are any specific instances where the citations appear insufficient or unclear, we would be grateful if the reviewer could indicate them, so we can address these points appropriately
- Most of the references are too old, and it is advisable to cite the latest literature from the past three years as much as possible.
Author´s response:
We acknowledge the reviewer’s comment and the old references were deleted from the manuscript or replaced by more recent ones.
- At the very least, authors should provide a detailed introduction to Silirum ®.
Author´s response:
We agree with the reviewer’s comment and a statement about Silirum® vaccine (strain, inactivation method, adjuvant) was included in the introduction.
REFERENCES
A Practical Information-Theoretic Approach. (2002). In Burnham, K.P., Anderson, D.R. (Eds.), Model selection and multimodel inference. Springer: New York.
Bafica, A., Scanga, C. A., Feng, C. G., Leifer, C., Cheever, A., & Sher, A. (2005). TLR9 regulates Th1 responses and cooperates with TLR2 in mediating optimal resistance to Mycobacterium tuberculosis. The Journal of experimental medicine, 202(12), 1715–1724. https://doi.org/10.1084/jem.20051782
Chang, E. Y., Guo, B., Doyle, S. E., & Cheng, G. (2007). Cutting edge: involvement of the type I IFN production and signaling pathway in lipopolysaccharide-induced IL-10 production. Journal of immunology (Baltimore, Md.: 1950), 178(11), 6705–6709. https://doi.org/10.4049/jimmunol.178.11.6705
Fernández, M., Benavides, J., Castaño, P., Elguezabal, N., Fuertes, M., Muñoz, M., Royo, M., Ferreras, M. C., & Pérez, V. (2017). Macrophage Subsets Within Granulomatous Intestinal Lesions in Bovine Paratuberculosis. Veterinary pathology, 54(1), 82–93. https://doi.org/10.1177/0300985816653794
Whittingham, M.J., Stephens, P.A., Bradbury, R.B., Freckleton, R.P. (2006). Why Do We Still Use Stepwise Modelling in Ecology and Behaviour? Journal of Animal Ecololy, 75, 1182–1189. https://doi.org/10.1111/J.1365-2656.2006.01141.X.
Zuur, A.F., Ieno, E.N., Elphick, C.S. (2010). A Protocol for Data Exploration to Avoid Common Statistical Problems. Methods in Ecology and Evolution, 1, 3–14. https://doi.org/10.1111/J.2041-210X.2009.00001.X.
Zuur, A.F., Ieno, E.N., Smith, G.M. (2007). Analysing Ecological Data. https://doi.org/10.1007/978-0-387-45972-1
Reviewer 4 Report
Comments and Suggestions for Authors
Dear Editor,
Thank you for the opportunity to review the manuscript entitled “Influence of paratuberculosis vaccination on the immunohistochemical expression of TLR1, TLR2, TLR4, TLR9, IFN-γ, iNOS 3 and CD204 in experimentally infected calves”.
My professional opinion is that this manuscript is relevant for the field of paratuberculosis, pathology, and immunology. However, it needs substantial modifications in the results and discussion, and an explanation of the methodology chosen by the authors.
My main concern is why there is immunolabelling for membranous TLRs in the cytoplasm of macrophages instead of the plasma membrane.
TLRs immunolabelling just reflects the presence of those receptors, and, as a reminder for the authors, immunohistochemistry is not a technique assessing protein expression, so the obtained semi-quantitative data should be contrasted with other techniques.
I disagree with the authors in that trained immunity occurs in this case with the current methodology – this is a major issue, and suggesting this repeatedly in the discussion based on speculations and questionable comparisons is not an adequate scientific practice.
If the authors have the intention of addressing the in-situ immune response, as stated, why has immunohistochemistry for T-cells, B-cells, plasma cells, etc. has not been performed? Where is the data of one of the best measurements for PTB immune response efficacy - bacillary load (ZN > FF > IHC > ISH)?
My comments for the authors are below, and I am looking forward to discussing them in the following round of revisions, if agreed by the Editor.
Major:
Immunohistochemistry visually highlights the labelling of an antibody to an epitope in a tissue section. “Expression” and similar wording are not adequate for immunohistochemistry unless correlated with other quantitative techniques. Please correct the terminology in the title, summary, and throughout the text.
I appreciate the introduction being concise and to the point. However, in L56-58, the authors mention trained immunity – these lines are out of context and do not align with the initial intention of the authors to evaluate the in situ immune response and the importance of TLRs in PTB. Please cancel these lines (L56-58) and discuss if your findings may be related to trained immunity, if able to prove it. I do not think trained immunity can be proven with the methodology applied in this study (see below).
L60: “Scant”. The authors should disclose what the “scant” knowledge is. Scant is subjective. If no information is available, then state: We found no studies focused on TLR labelling by immunohistochemistry in bovines in a search of several academic resources, including Google Scholar, PubMed, and CAD Direct, using search terms X, Y, Z”. What about in other domestic ruminants?
L71: We vaccinate (with more or less success) against pathogens, not diseases. Please amend throughout the text.
L73: “biological markers of innate immunity were not evaluated in these studies.” What do the authors mean? This is a vague statement. I think the introduction should focus on the actual knowledge, and the main gap to be filled – some statements out of context may be misinterpreted by the readers.
L80: I am concerned that this may be misinterpreted, as ethical approval is a sensitive issue. Please add the following: “The archived formalin-fixed embedded tissue (which tissues) samples of 16 Holstein calves from a previous study (ethical approval needed and cite if published) were used for this study.” To be clear, I am not concerned about ethical issues, but I would rather be transparent. The second sentence is not needed if replaced by the proposed.
L101-103: “These included sections with an isotype control for the primary antibody and the omission of the primary antibody”. Omission of the primary antibody is not sufficient for negative controls, see: https://pubmed.ncbi.nlm.nih.gov/24129895/. Please provide an example of the negative for each image you provided in the Supplementary Material. You could have used commercially available universal negative control reagent
L171: Should TLR immunolabeling be membranous instead…?
L205: That’s because CD204 is not present in all types of macrophages – did you consider using an alternative antibody such as IBA-1?
L207: CD204 cannot be stained because IHC is not a stain – it is an immunologic technique – amend elsewhere in the manuscript.
L209: I am more interested in knowing where the labelling for TLRs is instead of whether there is/not.
L2010: “Staining” > immunolabelling
L211: The only “cytoplasmic” (endosomal) TLRs are 3, 7, 8, 9, and 13… This looks like a nonspecific background in a hemosiderophage.
Result section: Please, detail where it is (at a cellular level) in each statement.
L491-493: By which method and where did Jeyanathan et al. report TLR2? So are you suggesting that just because there is increased TLR2 immunolabelling by IHC, is innate immunity in your vaccinated calves?
L494-495: The authors should first find a reference for TLR2 immunolabelling in the cytoplasm that can prove that it is real immunolabelling in their cases, and go from there. To demonstrate trained immunity by immunohistochemistry (IHC), one must show upregulation of innate immune markers in tissue macrophages or monocytes following a priming event. This involves labelling proinflammatory cytokines such as IL-1β, TNF-α, and IL-6, or other markers like CD68, CD163, or H3K4me3, and contrasting the usually not reliable cytokine IHC results with proteomics and/or transcriptomics. Unless the authors want to include this methodology from FFPE tissue, they cannot discuss that their results represent trained immunity.
The discussion section relies on TLR2 for the most part; however, the rest of findings, that may be useful (and in my opinion more accurate) to assess the local IR in PTB vaccinated calves are ignored for the most part – I suggest the authors to restructure the discussion and expand it in every aspect unrelated with TLR2. I lack cohesion on the decision on labelling macrophages, NOS, IFN and TLRs. This looks aleatory rather than a thoughtful decision as per the discussion.
The discussion should compare and contrast the results with published data. I do not see discussion but repetitive statements from the results and vague, weak associations with studies without disclosing methodologies in those studies (i.e., it is not clear if the studies use IHC, proteomics, or another technique). I believe that the authors can work on the discussion to integrate information and achieve a minimum standard according to the originality of the idea and the laborious laboratory work.
Minor:
L47: I would suggest adding Johne disease as a synonym here as it is a popular term. I personally do not use eponymous – up to the authors.
L62-70: The information about IFN and M1/M2 requires an independent paragraph
L70-77: This is to me one paragraph
L85-87: Pathologists use continuously archived material for pathogenesis studies. No need to disclose except if the material is used repeatedly.
L87: ”Those samples selected” - Which? How many animals?
L89: The readers should have the methodology available from the current manuscript (González et al. [34]”. I do not agree with “personification” in citations.
L89-92: Please expand this information.
L100: IHC is a procedure - please state “Immunohistochemistry” instead of “immunohistochemistry procedure”. There are thousands of IHC protocols – please clarify what was used in ref. [32]
L101: Negative controls?
L109-111: 30 counts in each compartment on 10 in each? It is not clear from the text.
L112-114: So, the inoculation site and scapular lymph nodes were also included? As per the nomina anatomic veterinaria, there is no prescapular lymph node.
L107-119 should be reorganized for clarity – this paragraph is dense and somehow chaotic.
L123-124: The methodology should be self-explanatory – the readers should be consulting 10 manuscripts while reading yours - what’s H-score?
L135-137: Who (initials) performed the separate evaluation, and who (initials) participated in the consensus score? Just to be clear, state veterinary pathologists/anatomic pathologists in training/boarded (if applicable), so that it is emphasized that the core of the analysis was done and led by veterinary pathologists, and not by other random scientists.
L138: Is this section necessary? The analysis is already rigorous; if this information needs to be disclosed by the authors, please move it to the Suppl. Material and state essential tests are used for analysis.
L172-175: As per the rigour of this manuscript, I am sure that the background of the authors allows them to state macrophage, lymphocyte, and neutrophil. Name them accordingly. If a cell is labelled CD204, it will be a macrophage – please be concise. Did macrophages express IFN?
L198: “Similar distribution to controls”
L486-488: This is obvious as these were not biopsies – please cancel.
L490: Immunolabelling + the epitope (not for the epitope)
L507: I am not sure that the pattern recognition receptor has been defined before using an abbreviation
L515-517: How can a TLR be related to cytokine production?
L521: It is “probably key”?
Comments on the Quality of English LanguageEnglish revision may be needed to increase flow after scientific issues have been adressed.
Author Response
First of all, we would like to thank the editor and reviewers for their additional comments and suggestions that have contributed to improve even more the quality of the first revision. We have carefully revised the manuscript and addressed all the comments and corrections suggested.
Major changes:
- Immunohistochemistry visually highlights the labelling of an antibody to an epitope in a tissue section. “Expression” and similar wording are not adequate for immunohistochemistry unless correlated with other quantitative techniques. Please correct the terminology in the title, summary, and throughout the text.
Author´s response:
Agreed. We acknowledge the reviewer recommendation and the manuscript was modified accordingly.
- I appreciate the introduction being concise and to the point. However, in L56-58, the authors mention trained immunity – these lines are out of context and do not align with the initial intention of the authors to evaluate the in situ immune response and the importance of TLRs in PTB. Please cancel these lines (L56-58) and discuss if your findings may be related to trained immunity, if able to prove it. I do not think trained immunity can be proven with the methodology applied in this study (see below).
Author´s response:
We agree with the reviewer’s comment. The section of the introduction referring to innate immune training has been removed and replaced with a statement focusing specifically on Toll-like receptors, which better aligns with the scope and methodology of our study.
- L60: “Scant”. The authors should disclose what the “scant” knowledge is. Scant is subjective. If no information is available, then state: We found no studies focused on TLR labelling by immunohistochemistry in bovines in a search of several academic resources, including Google Scholar, PubMed, and CAD Direct, using search terms X, Y, Z”. What about in other domestic ruminants?
Author´s response:
To the best of our knowledge, only two publications address the local expression of Toll-like receptors (TLRs) at the site of Mycobacterium avium subsp. paratuberculosis (Map) infection in cattle (Subharat et al., 2012; Zapico et al., 2025), one of which was recently published by our group (Zapico et al., 2025). This contrasts with the relatively large body of literature on the relationship between single nucleotide polymorphisms (SNPs) in TLR genes and susceptibility to bovine paratuberculosis (Kravitz et al., 2021). Regarding other domestic ruminants, such as sheep, we found three publications on the local expression of TLRs (Nalubamba et al., 2008; Plain et al., 2010; Taylor et al., 2008). However, these were not included in the introduction since our study focuses specifically on cattle. Following the reviewer’s recommendation, the phrase “…but the information about their expression at the infection site is scant” was revised to “…but there are fewer studies about their expression at the infection site.”
- L71: We vaccinate (with more or less success) against pathogens, not diseases. Please amend throughout the text.
Author´s response:
Agreed. The text was modified according to the reviewer comment.
- L73: “biological markers of innate immunity were not evaluated in these studies.” What do the authors mean? This is a vague statement. I think the introduction should focus on the actual knowledge, and the main gap to be filled – some statements out of context may be misinterpreted by the readers.
Author´s response:
We agree with the reviewer’s observation. The statement regarding “biological markers of innate immunity were not evaluated in these studies” was deemed vague and has been removed from the manuscript to improve clarity and focus on the relevant knowledge gap.
- L80: I am concerned that this may be misinterpreted, as ethical approval is a sensitive issue. Please add the following: “The archived formalin-fixed embedded tissue (which tissues) samples of 16 Holstein calves from a previous study (ethical approval needed and cite if published) were used for this study.” To be clear, I am not concerned about ethical issues, but I would rather be transparent. The second sentence is not needed if replaced by the proposed.
Author´s response:
We agree with the reviewer’s suggestion to improve transparency regarding ethical approval. The ethical approval section was revised to include the following statement: “Archived formalin-fixed embedded tissue samples from 16 Holstein calves from a previous study were used. The original study, a PhD thesis conducted before current ethical guidelines were established, did not include a specific ethical approval statement, which we have noted in the manuscript.” The second sentence originally included was removed as per the reviewer’s recommendation.
- L101-103: “These included sections with an isotype control for the primary antibody and the omission of the primary antibody”. Omission of the primary antibody is not sufficient for negative controls, see: https://pubmed.ncbi.nlm.nih.gov/24129895/. Please provide an example of the negative for each image you provided in the Supplementary Material. You could have used commercially available universal negative control reagent
Author´s response:
We appreciate the reviewer’s reference. The cited article acknowledges that omission of the primary antibody, while considered an inferior option, is an accepted form of negative control (page 70, last sentence of the first paragraph). In our study, the specificity of the primary antibodies and the absence of cross-reactivity were confirmed through three approaches: Western blot analysis of stimulated bovine leukocytes, omission of the primary antibody substituted by antibody diluent, and use of species- and isotype-matched immunoglobulins as controls. The species- and isotype-matched immunoglobulins were generated in-house and purified from mouse and rabbit serum, respectively. Additionally, examples of negative controls corresponding to each image are provided in the Supplementary Material.
- L171: Should TLR immunolabeling be membranous instead…?
Author´s response:
As the reviewer suggests, TLR1, TLR2, and TLR4 are typically expressed on the cell surface, so a membranous staining pattern would be expected. However, in our study, immunolabeled cells consistently showed cytoplasmic staining. This observation aligns with previous findings from our group (Zapico et al., 2025) and other researchers (Chen et al., 2020; Ng et al., 2011; Han et al., 2022; Kasurinen et al., 2019). One possible explanation is that the antigen retrieval process may cause solubilization or redistribution of membrane proteins into the cytoplasm. Additionally, since surface TLRs are synthesized in the endoplasmic reticulum and transported through the Golgi apparatus to the plasma membrane via the secretory pathway, the cytoplasmic staining may reflect labeling of TLR proteins in the process of synthesis within these organelles.
- L205: That’s because CD204 is not present in all types of macrophages – did you consider using an alternative antibody such as IBA-1?
Author´s response:
We appreciate the reviewer’s suggestion. One of the primary aims of this study was to characterize M1/M2 macrophage polarization within granulomas, particularly to assess whether vaccination influenced macrophage phenotypes in MAP-associated lesions. For this reason, we chose CD204, a marker associated with M2 macrophages, rather than a general macrophage marker such as IBA-1, which would not differentiate between macrophage subsets and therefore would not contribute additional insight into polarization. The observation that CD204⁺ macrophages were predominantly localized at the periphery of granulomas in vaccinated animals was emphasized here, as this spatial distribution is later discussed in the context of its potential role in limiting inflammation and contributing to granuloma latency.
- L207: CD204 cannot be stained because IHC is not a stain – it is an immunologic technique – amend elsewhere in the manuscript.
Author´s response:
Agreed. The manuscript was modified accordingly.
- L209: I am more interested in knowing where the labelling for TLRs is instead of whether there is/not.
Author´s response:
Agreed. The distribution of the immunolabeling for the different markers at cell level was included in the result section.
- L2010: “Staining” > immunolabelling
Author´s response:
Agreed.
- L211: The only “cytoplasmic” (endosomal) TLRs are 3, 7, 8, 9, and 13… This looks like a nonspecific background in a hemosiderophage.
Author´s response:
We understand the reviewer’s concern regarding the TLR4 immunolabeling observed in granulomatous lesions. While TLR4 is primarily a cell surface receptor, it is known to undergo internalization into endosomes following activation (Ciesielska et al., 2021). In our study, the majority of macrophages within multifocal and diffuse granulomatous lesions exhibited a distinctive dot-like cytoplasmic immunolabeling pattern. Importantly, these cells did not show evidence of pigment accumulation (e.g., hemosiderin or lipofuscin) on hematoxylin-eosin-stained sections, and no similar labeling was seen in negative controls. Furthermore, the specificity of the TLR4 primary antibody was confirmed via Western blot analysis. Therefore, we consider the observed staining to be specific and likely indicative of TLR4-containing endosomes, as discussed in the manuscript. A similar staining pattern has also been reported in MAP-associated diffuse multibacillary lesions with high bacterial loads in our previous study (Zapico et al., 2025).
- Result section: Please, detail where it is (at a cellular level) in each statement.
Author´s response:
Agreed. The distribution of the immunolabeling along the cell for the different markers was described along the result section.
- L491-493: By which method and where did Jeyanathan et al. report TLR2? So are you suggesting that just because there is increased TLR2 immunolabelling by IHC, is innate immunity in your vaccinated calves?
Author´s response:
We thank the reviewer for this important observation. Jeyanathan et al. (2022) investigated TLR2 expression by isolating alveolar macrophages from BCG-vaccinated mice and analyzing them via flow cytometry after ex vivo stimulation with BCG. We acknowledge that our study, which relies on immunohistochemistry, cannot directly demonstrate innate immune training or make mechanistic conclusions about TLR2 upregulation in vaccinated calves. We also agree with the reviewer that parts of the discussion may have overemphasized this speculative hypothesis. In response, we have revised the discussion to focus more on the actual findings of the study and expanded sections that are more directly related to our results.
- L494-495: The authors should first find a reference for TLR2 immunolabelling in the cytoplasm that can prove that it is real immunolabelling in their cases, and go from there. To demonstrate trained immunity by immunohistochemistry (IHC), one must show upregulation of innate immune markers in tissue macrophages or monocytes following a priming event. This involves labelling proinflammatory cytokines such as IL-1β, TNF-α, and IL-6, or other markers like CD68, CD163, or H3K4me3, and contrasting the usually not reliable cytokine IHC results with proteomics and/or transcriptomics. Unless the authors want to include this methodology from FFPE tissue, they cannot discuss that their results represent trained immunity.
Author´s response:
We appreciate the reviewer’s detailed comments and fully agree that trained immunity cannot be demonstrated using the methodology employed in our study. As such, the discussion sections related to this speculative interpretation have been removed. Regarding TLR2 immunolabeling, we note that cytoplasmic staining has been previously reported in both inflammatory (Chen et al., 2020; Ng et al., 2011) and neoplastic cells (Kasurinen et al., 2019). As mentioned in our response to Comment 8, this pattern may result from solubilization of membrane proteins during antigen retrieval or reflect intracellular localization of TLR2 during its synthesis and trafficking through the endoplasmic reticulum and Golgi apparatus
- The discussion section relies on TLR2 for the most part; however, the rest of findings, that may be useful (and in my opinion more accurate) to assess the local IR in PTB vaccinated calves are ignored for the most part – I suggest the authors to restructure the discussion and expand it in every aspect unrelated with TLR2. I lack cohesion on the decision on labelling macrophages, NOS, IFN and TLRs. This looks aleatory rather than a thoughtful decision as per the discussion.
Author´s response:
We appreciate the reviewer’s insightful comment. We agree that the original discussion placed disproportionate emphasis on TLR2, potentially overlooking other important findings related to the local immune response in vaccinated calves. In response, we have restructured and expanded the discussion to better reflect the interplay between innate immunity (including TLRs) and cell-mediated immunity (e.g., IFN-γ and iNOS expression). We have also emphasized additional findings, such as the immune response observed at the vaccine granuloma site, to provide a more balanced and integrated interpretation of our results. Furthermore, we clarified the rationale behind the selection of markers for immunolabeling to demonstrate that these were chosen based on their established roles in MAP-associated immune responses.
- The discussion should compare and contrast the results with published data. I do not see discussion but repetitive statements from the results and vague, weak associations with studies without disclosing methodologies in those studies (i.e., it is not clear if the studies use IHC, proteomics, or another technique). I believe that the authors can work on the discussion to integrate information and achieve a minimum standard according to the originality of the idea and the laborious laboratory work.
Author´s response:
We appreciate the reviewer’s comment and fully agree that the discussion required improvement to more effectively contextualize our findings within the existing literature. In the revised manuscript, we have restructured the discussion to move beyond repetition of the results and instead focus on comparing and contrasting our findings with those of previous studies. We have clearly indicated the methodologies used in the cited works (e.g., immunohistochemistry, in vitro stimulation, transcriptomics, or proteomics) to ensure that the nature and limitations of each comparison are transparent.
We would also like to note that much of the available literature on the innate immune response to MAP—particularly in the context of vaccination—has been conducted using in vitro models. There remains a significant gap in in vivo studies evaluating immune responses in the intestinal tissue of vaccinated animals. Despite this limitation, we have aimed to integrate our findings with the most relevant published data and to highlight both consistencies and discrepancies, thereby better supporting the originality and significance of our study.
Minor:
- L47: I would suggest adding Johne disease as a synonym here as it is a popular term. I personally do not use eponymous – up to the authors.
Author´s response:
Agreed.
- L62-70: The information about IFN and M1/M2 requires an independent paragraph
Author´s response:
Agreed.
- L70-77: This is to me one paragraph
Author´s response:
Agreed.
- L85-87: Pathologists use continuously archived material for pathogenesis studies. No need to disclose except if the material is used repeatedly.
Author´s response:
Agreed.
- L87: ”Those samples selected” - Which? How many animals?
Author´s response:
Agreed. The precise tissue samples employed were described in the manuscript.
- L89: The readers should have the methodology available from the current manuscript (González et al. [34]”. I do not agree with “personification” in citations.
Author´s response:
Agreed. According to the reviewer’s comment, a brief description of each type of lesion was included in the text.
- L89-92: Please expand this information.
Author´s response:
Agreed. This section was expanded indicating the number of animals in each group for better clarity.
- L100: IHC is a procedure - please state “Immunohistochemistry” instead of “immunohistochemistry procedure”. There are thousands of IHC protocols – please clarify what was used in ref. [32]
Author´s response:
Agreed. A brief description of the protocol was added.
- L101: Negative controls?
Author´s response:
Agreed.
- L109-111: 30 counts in each compartment on 10 in each? It is not clear from the text.
Author´s response:
The cell counting was performed in 30 fields of each location. The text was modified for better clarity.
- L112-114: So, the inoculation site and scapular lymph nodes were also included? As per the nomina anatomic veterinaria, there is no prescapular lymph node.
Author´s response:
The inoculation site and scapular lymph nodes were analyzed like the other intestinal locations (LP, GALT and LN). According to the reviewer comment, the term “prescapular lymph node (PLN)” was changed to “scapular lymph node (SLN)” along the text for a more rigorous anatomic terminology.
- L107-119 should be reorganized for clarity – this paragraph is dense and somehow chaotic.
Author´s response:
Agreed. The paragraph was modified for better clarity.
- L123-124: The methodology should be self-explanatory – the readers should be consulting 10 manuscripts while reading yours - what’s H-score?
Author´s response:
Agreed. A brief description of the H-score method for evaluating IHC was added in the text.
- L135-137: Who (initials) performed the separate evaluation, and who (initials) participated in the consensus score? Just to be clear, state veterinary pathologists/anatomic pathologists in training/boarded (if applicable), so that it is emphasized that the core of the analysis was done and led by veterinary pathologists, and not by other random scientists.
Author´s response:
Agreed.
- L138: Is this section necessary? The analysis is already rigorous; if this information needs to be disclosed by the authors, please move it to the Suppl. Material and state essential tests are used for analysis.
Author´s response:
We thank the reviewer for this comment. While we understand the suggestion to move the statistical analysis section to the Supplementary Material, we respectfully believe it is appropriate to retain it in the main Methods section. Clear and transparent reporting of statistical methods is essential for the reproducibility and critical evaluation of the study. Although the analysis is rigorous, explicitly including the statistical approach in the main text allows readers to fully understand the rationale behind the analyses without needing to refer to supplementary content. For these reasons, we prefer to retain this section as it currently stands.
- L172-175: As per the rigour of this manuscript, I am sure that the background of the authors allows them to state macrophage, lymphocyte, and neutrophil. Name them accordingly. If a cell is labelled CD204, it will be a macrophage – please be concise. Did macrophages express IFN?
Author´s response:
Agreed. The paragraph was summarized according to the reviewer’s comment. In the present work, no IFN-y immunolabeling was observed in the macrophages associated or unrelated to the granulomas.
- L198: “Similar distribution to controls”
Author´s response:
Agreed.
- L486-488: This is obvious as these were not biopsies – please cancel.
Author´s response:
Agreed.
- L490: Immunolabelling + the epitope (not for the epitope)
Author´s response:
Agreed.
- L507: I am not sure that the pattern recognition receptor has been defined before using an abbreviation
Author´s response:
We acknowledge the reviewer comment. The introduction was modified according to the previous reviewer’s comments and now the term “pattern recognition receptor (PRR)” now appears in the introduction.
- L515-517: How can a TLR be related to cytokine production?
Author´s response:
Studies such as Bafica et al. (2005) have shown that activation of TLR9 by mycobacterial DNA in murine macrophages induces the production of IL-12 in a TLR9-dependent manner. Based on this, we suggest that in our system, activation of TLR9 by MAP DNA—potentially released from lysed bacteria within macrophages—may trigger IL-12 secretion by these cells, thereby supporting the maintenance of an effective Th1 immune response. We acknowledge that the original wording in lines 515–517 may have been unclear, so these lines were revised for improved clarity.
- L521: It is “probably key”?
Author´s response:
Agreed.
REFERENCES
Bafica, A., Scanga, C. A., Feng, C. G., Leifer, C., Cheever, A., & Sher, A. (2005). TLR9 regulates Th1 responses and cooperates with TLR2 in mediating optimal resistance to Mycobacterium tuberculosis. The Journal of experimental medicine, 202(12), 1715–1724. https://doi.org/10.1084/jem.20051782
Chen, X., Zhao, D., Ning, Y., & Zhou, Y. (2020). Toll-like receptors 2 expression in mediastinal lymph node of patients with sarcoidosis. Annals of translational medicine, 8(18), 1182. https://doi.org/10.21037/atm-20-6103
Ciesielska, A., Matyjek, M., & Kwiatkowska, K. (2021). TLR4 and CD14 trafficking and its influence on LPS-induced pro-inflammatory signaling. Cellular and molecular life sciences: CMLS, 78(4), 1233–1261. https://doi.org/10.1007/s00018-020-03656-y
Gupta, S. K., Wilson, T., Maclean, P. H., Rehm, B. H. A., Heiser, A., Buddle, B. M., & Wedlock, D. N. (2023). Mycobacterium avium subsp. paratuberculosis antigens induce cellular immune responses in cattle without causing reactivity to tuberculin in the tuberculosis skin test. Frontiers in immunology, 13, 1087015. https://doi.org/10.3389/fimmu.2022.1087015
Han, J. H., Ahn, M. H., Jung, J. Y., Kim, J. W., Suh, C. H., Kwon, J. E., Yim, H., & Kim, H. A. (2022). Elevated expression of TLR2 and its correlation with disease activity and clinical manifestations in adult-onset Still's disease. Scientific reports, 12(1), 10240. https://doi.org/10.1038/s41598-022-14004-4
Jeyanathan, M., Vaseghi-Shanjani, M., Afkhami, S., Grondin, J. A., Kang, A., D'Agostino, M. R., Yao, Y., Jain, S., Zganiacz, A., Kroezen, Z., Shanmuganathan, M., Singh, R., Dvorkin-Gheva, A., Britz-McKibbin, P., Khan, W. I., & Xing, Z. (2022). Parenteral BCG vaccine induces lung-resident memory macrophages and trained immunity via the gut-lung axis. Nature immunology, 23(12), 1687–1702. https://doi.org/10.1038/s41590-022-01354-4
Kasurinen, A., Hagström, J., Laitinen, A., Kokkola, A., Böckelman, C., & Haglund, C. (2019). Evaluation of toll-like receptors as prognostic biomarkers in gastric cancer: high tissue TLR5 predicts a better outcome. Scientific reports, 9(1), 12553. https://doi.org/10.1038/s41598-019-49111-2
Kleinnijenhuis, J., Quintin, J., Preijers, F., Benn, C. S., Joosten, L. A., Jacobs, C., van Loenhout, J., Xavier, R. J., Aaby, P., van der Meer, J. W., van Crevel, R., & Netea, M. G. (2014). Long-lasting effects of BCG vaccination on both heterologous Th1/Th17 responses and innate trained immunity. Journal of innate immunity, 6(2), 152–158. https://doi.org/10.1159/000355628
Kravitz, A., Pelzer, K., & Sriranganathan, N. (2021). The Paratuberculosis Paradigm Examined: A Review of Host Genetic Resistance and Innate Immune Fitness in Mycobacterium avium subsp. Paratuberculosis Infection. Frontiers in veterinary science, 8, 721706. https://doi.org/10.3389/fvets.2021.721706
Nalubamba, K., Smeed, J., Gossner, A., Watkins, C., Dalziel, R., & Hopkins, J. (2008). Differential expression of pattern recognition receptors in the three pathological forms of sheep paratuberculosis. Microbes and infection, 10(6), 598–604. https://doi.org/10.1016/j.micinf.2008.02.005
Ng, L. K., Rich, A. M., Hussaini, H. M., Thomson, W. M., Fisher, A. L., Horne, L. S., & Seymour, G. J. (2011). Toll-like receptor 2 is present in the microenvironment of oral squamous cell carcinoma. British journal of cancer, 104(3), 460–463. https://doi.org/10.1038/sj.bjc.6606057
Plain, K. M., Purdie, A. C., Begg, D. J., de Silva, K., & Whittington, R. J. (2010). Toll-like receptor (TLR)6 and TLR1 differentiation in gene expression studies of Johne's disease. Veterinary immunology and immunopathology, 137(1-2), 142–148. https://doi.org/10.1016/j.vetimm.2010.04.002
Subharat, S., Shu, D., de Lisle, G. W., Buddle, B. M., & Wedlock, D. N. (2012). Altered patterns of toll-like receptor gene expression in cull cows infected with Mycobacterium avium subsp. paratuberculosis. Veterinary immunology and immunopathology, 145(1-2), 471–478. https://doi.org/10.1016/j.vetimm.2011.10.008
Taylor, D. L., Zhong, L., Begg, D. J., de Silva, K., & Whittington, R. J. (2008). Toll-like receptor genes are differentially expressed at the sites of infection during the progression of Johne's disease in outbred sheep. Veterinary immunology and immunopathology, 124(1-2), 132–151. https://doi.org/10.1016/j.vetimm.2008.02.021
Zapico, D., Espinosa, J., Criado, M., Gutiérrez, D., Ferreras, M. D. C., Benavides, J., Pérez, V., & Fernández, M. (2025). Immunohistochemical expression of TLR1, TLR2, TLR4, and TLR9 in the different types of lesions associated with bovine paratuberculosis. Veterinary pathology, 62(3), 305–318. https://doi.org/10.1177/03009858241302850
Round 2
Reviewer 3 Report
Comments and Suggestions for Authors
- Why are the scar bars used differently for each group in Figure 1& Figure 2? 20um, 30um, 50um, 100um. However, in Figures 3 and 4, 100um and 50um are respectively used?
- The author aims to evaluate TLR1, TLR2, TLR4, TLR9, IFN - γ, iNOS, and CD204 vaccination with Silirum ® What are the selection criteria for these genes in calves experimentally infected with paratuberculosis after vaccination? Why choose these proteins from the TLR family?
- The modified parts should be marked with colors in the revised draft for better differentiation.
Author Response
First of all, we would like to thank the reviewer for the additional comments and suggestions that have contributed to improve even more the quality of the first revision.
- Why are the scar bars used differently for each group in Figure 1& Figure 2? 20um, 30um, 50um, 100um. However, in Figures 3 and 4, 100um and 50um are respectively used?
Author’s response:
In Figures 3 and 4, the scale bars are consistent in length because the images within each figure were captured at the same magnification (100x for images in Figure 3 and 200x for those in Figure 4). In contrast, Figures 1 and 2 contain microphotographs taken at varying magnifications to highlight different features. For example, higher magnifications were used to better visualize intracellular staining patterns (e.g., Figure 1L, Figure 2O–P), while lower magnifications were used to show spatial distribution within lesions or across tissue sections (e.g., Figure 1A, Figure 2E, 2L). This naturally results in scale bars representing different actual distances (100, 50, 30, or 20 μm). Nevertheless, these distances were carefully chosen to maintain visual consistency, so the scale bars appear similar in size across panels despite reflecting different spatial scales. This approach allows accurate interpretation of image dimensions while preserving both clarity and consistency.
- The author aims to evaluate TLR1, TLR2, TLR4, TLR9, IFN - γ, iNOS, and CD204 vaccination with Silirum ® What are the selection criteria for these genes in calves experimentally infected with paratuberculosis after vaccination? Why choose these proteins from the TLR family?
Author’s response:
According to the literature, TLR1, TLR2, TLR4, and TLR9 are the main Toll-like receptors involved in the recognition of mycobacteria by innate immune cells and the onset of the immune response (Kim et al., 2019; Mehta et al., 2021; Vu et al., 2017). Specifically, TLR1, TLR2, TLR4 and TLR9 have been directly associated with MAP infection, since SNPs in the corresponding TLR genes are linked to increased susceptibility to paratuberculosis in cattle (Kravitz et al., 2021). These TLRs are also known to initiate signaling pathways leading the onset of IFN-γ dependent Th1 immune response, which is key for M1 macrophage activation and a successful control of mycobacteria (Bafica et al., 2005; Shapouri-Moghaddam et al., 2018).
In previous immunohistochemical studies conducted by our group (Fernandez et al., 2017a, 2017b; Zapico et al., 2025), we observed differences in the expression of TLR1, TLR2, TLR4, TLR9, IFN-γ, M1 and M2 macrophage markers in the intestine of naturally infected animals depending on the type and extension of the granulomatous lesions. These findings suggest an important role of these markers in the pathogenesis of the disease. Based on this background, we selected these specific markers to evaluate whether vaccination with Silirum® modulates their expression in calves experimentally infected with MAP, thus providing an insight into the immune mechanisms underlying the protection provided by this vaccine.
- The modified parts should be marked with colors in the revised draft for better differentiation.
Author’s response:
We appreciate your comment. In response, we have uploaded two versions of the revised manuscript: one clean version and one with the changes clearly marked using tracked changes, as requested. Please let us know if any additional formatting is needed.
REFERENCES
Bafica, A., Scanga, C. A., Feng, C. G., Leifer, C., Cheever, A., & Sher, A. (2005). TLR9 regulates Th1 responses and cooperates with TLR2 in mediating optimal resistance to Mycobacterium tuberculosis. The Journal of experimental medicine, 202(12), 1715–1724.
Fernández, M., Benavides, J., Castaño, P., Elguezabal, N., Fuertes, M., Muñoz, M., Royo, M., Ferreras, M. C., & Pérez, V. (2017a). Macrophage Subsets Within Granulomatous Intestinal Lesions in Bovine Paratuberculosis. Veterinary pathology, 54(1), 82–93.
Fernández, M., Fuertes, M., Elguezabal, N., Castaño, P., Royo, M., Ferreras, M. C., Benavides, J., & Pérez, V. (2017b). Immunohistochemical expression of interferon-γ in different types of granulomatous lesions associated with bovine paratuberculosis. Comparative immunology, microbiology and infectious diseases, 51, 1–8.
Kim, J. S., Kim, Y. R., y Yang, C. S. (2019). Latest comprehensive knowledge of the crosstalk between tlr signaling and mycobacteria and the antigens driving the process. Journal of microbiology and biotechnology, 29(10), 1506–1521.
Kravitz, A., Pelzer, K., & Sriranganathan, N. (2021). The Paratuberculosis Paradigm Examined: A Review of Host Genetic Resistance and Innate Immune Fitness in Mycobacterium avium subsp. Paratuberculosis Infection. Frontiers in veterinary science, 8, 721706.
Mehta, P., Ray, A., y Mazumder, S. (2021). TLRs in mycobacterial pathogenesis: black and white or shades of gray. Current microbiology, 78(6), 2183–2193.
Shapouri-Moghaddam, A., Mohammadian, S., Vazini, H., Taghadosi, M., Esmaeili, S. A., Mardani, F., Seifi, B., Mohammadi, A., Afshari, J. T., & Sahebkar, A. (2018). Macrophage plasticity, polarization, and function in health and disease. Journal of cellular physiology, 233(9), 6425–6440.
Vu, A., Calzadilla, A., Gidfar, S., Calderon-Candelario, R., y Mirsaeidi, M. (2017). Toll-like receptors in mycobacterial infection. European journal of pharmacology, 808, 1–7.
Zapico, D., Espinosa, J., Criado, M., Gutiérrez, D., Ferreras, M. D. C., Benavides, J., Pérez, V., & Fernández, M. (2025). Immunohistochemical expression of TLR1, TLR2, TLR4, and TLR9 in the different types of lesions associated with bovine paratuberculosis. Veterinary pathology, 62(3), 305–318.
Reviewer 4 Report
Comments and Suggestions for Authors
Dear Editor,
I appreciate the discussion with the authors and the learning opportunity while reviewing this manuscript. The authors have addressed all my previous concerns. In my opinion, this paper addresses significant gaps in the knowledge of paratuberculosis, and I recommend its publication after addressing the minor changes proposed below.
If the phagosome internalizes MAP along with TLRs, that may be why there is some degree of cytoplasmic immunolabelling in potentially endosomal vesicles.
Major:
“immunostaining” and “immunoreaction/reactivity/immunoreactivity” are still used in the text
A: As the reviewer suggests, TLR1, TLR2, and TLR4 are typically expressed on the cell surface, so a membranous staining pattern would be expected. However, in our study, immunolabeled cells consistently showed cytoplasmic staining. This observation aligns with previous findings from our group (Zapico et al., 2025) and other researchers (Chen et al., 2020; Ng et al., 2011; Han et al., 2022; Kasurinen et al., 2019). One possible explanation is that the antigen retrieval process may cause solubilization or redistribution of membrane proteins into the cytoplasm. Additionally, since surface TLRs are synthesized in the endoplasmic reticulum and transported through the Golgi apparatus to the plasma membrane via the secretory pathway, the cytoplasmic staining may reflect labeling of TLR proteins in the process of synthesis within these organelles.
R: Please integrate this into the discussion after L503 and combine with below
A: We understand the reviewer’s concern regarding the TLR4 immunolabeling observed in granulomatous lesions. While TLR4 is primarily a cell surface receptor, it is known to undergo internalization into endosomes following activation (Ciesielska et al., 2021). In our study, the majority of macrophages within multifocal and diffuse granulomatous lesions exhibited a distinctive dot-like cytoplasmic immunolabeling pattern. Importantly, these cells did not show evidence of pigment accumulation (e.g., hemosiderin or lipofuscin) on hematoxylin-eosin-stained sections, and no similar labeling was seen in negative controls. Furthermore, the specificity of the TLR4 primary antibody was confirmed via Western blot analysis. Therefore, we consider the observed staining to be specific and likely indicative of TLR4-containing endosomes, as discussed in the manuscript. A similar staining pattern has also been reported in MAP-associated diffuse multibacillary lesions with high bacterial loads in our previous study (Zapico et al., 2025).
R: Please integrate this into the discussion after L503 and combine with above
A: We thank the reviewer for this important observation. Jeyanathan et al. (2022) investigated TLR2 expression by isolating alveolar macrophages from BCG-vaccinated mice and analyzing them via flow cytometry after ex vivo stimulation with BCG.
R: Please define this when appropriate.
Minor changes:
L28 “of Toll-like receptor (TLR)-1, TLR2…”, same for IFN, iNOS, and CD – and please define the first time in the text
L31 takes > studied
L33 immunoreactivity > immunolabelling
L34 H-score – please define for the first time
L32 all lymphocytic? I think this is a typo
L32 immunolabelled TLR2
L36 immunoreactivity for > immunolabelling of
L40 in the intestine
L46 chronic wasting disease is a prionic disease of deer, please find another way to briefly define PTB… chronic enteritis? Up to the authors
L82-83: Cancel “The tissue samples used in this study came from animals used in an old study for 82 other purposes [7].” Start the sentence as “Archived formalin-fixed…”
L85: Cancel “(ethical approval not specified)” and just add “[7]”
L82-92: Repeated, cancel
L92 Briefly, those samples selected > Selected samples were categorized
L490-503: Fuse
L497 dpv should be DPV as it is an abbreviation – correct along the text
L518-521 please cancel, this is obvious
L518, 522, 551: lymphocytes/neutrophils immunolabelling IFN – you are not addressing production
L522: On the other hand – Cancel
L524: “In fact” – Cancel
L535: immunoreactivity > immunolabelling
L536: In the present work > Here
L540: signs > paratuberculosis
L540: Taken together – Cancel
L544-545: In this sense, a weak > Early deficient innate…
L551 or phagocytosis…
Other: “n” should be italicized
Comments on the Quality of English LanguageWould benefit from review by a native English speaker
Author Response
First of all, we would like to thank the reviewer for the additional comments and suggestions that have contributed to improve even more the quality of the first revision.
- If the phagosome internalizes MAP along with TLRs, that may be why there is some degree of cytoplasmic immunolabelling in potentially endosomal vesicles.
Author’s response:
In the case of TLR4 immunolabeling in MAP-associated multifocal and diffuse lesions, that is our hypothesis. We think that the dots observed in the cytoplasm of the macrophages forming these lesions are TLR4-containing endosomes associated with bacterial phagocytosis. This hypothesis is mentioned in the discussion section and in a previous immunohistochemical study performed by our group using intestinal tissue samples from naturally infected cattle (Zapico et al., 2025).
Major:
- “immunostaining” and “immunoreaction/reactivity/immunoreactivity” are still used in the text
Author’s response:
We acknowledge the reviewer’s comment and these terms were addressed in the manuscript.
- A:As the reviewer suggests, TLR1, TLR2, and TLR4 are typically expressed on the cell surface, so a membranous staining pattern would be expected. However, in our study, immunolabeled cells consistently showed cytoplasmic staining. This observation aligns with previous findings from our group (Zapico et al., 2025) and other researchers (Chen et al., 2020; Ng et al., 2011; Han et al., 2022; Kasurinen et al., 2019). One possible explanation is that the antigen retrieval process may cause solubilization or redistribution of membrane proteins into the cytoplasm. Additionally, since surface TLRs are synthesized in the endoplasmic reticulum and transported through the Golgi apparatus to the plasma membrane via the secretory pathway, the cytoplasmic staining may reflect labeling of TLR proteins in the process of synthesis within these organelles.
R: Please integrate this into the discussion after L503 and combine with below
Author’s response:
Agreed.
- A: We understand the reviewer’s concern regarding the TLR4 immunolabeling observed in granulomatous lesions. While TLR4 is primarily a cell surface receptor, it is known to undergo internalization into endosomes following activation (Ciesielska et al., 2021). In our study, the majority of macrophages within multifocal and diffuse granulomatous lesions exhibited a distinctive dot-like cytoplasmic immunolabeling pattern. Importantly, these cells did not show evidence of pigment accumulation (e.g., hemosiderin or lipofuscin) on hematoxylin-eosin-stained sections, and no similar labeling was seen in negative controls. Furthermore, the specificity of the TLR4 primary antibody was confirmed via Western blot analysis. Therefore, we consider the observed staining to be specific and likely indicative of TLR4-containing endosomes, as discussed in the manuscript. A similar staining pattern has also been reported in MAP-associated diffuse multibacillary lesions with high bacterial loads in our previous study (Zapico et al., 2025).
R: Please integrate this into the discussion after L503 and combine with above
Author’s response:
Agreed. In the new integrated paragraph, we mention that TLR4 dot-like immunolabeling in MAP-associated multifocal and diffuse lesions could represent TLR4-containing endosomes due to TLR4 activation in these lesions. However, we chose not to add here that this may also be related to the phagocytosis of MAP bacilli, as this is described in detail later in the discussion.
- A: We thank the reviewer for this important observation. Jeyanathan et al. (2022) investigated TLR2 expression by isolating alveolar macrophages from BCG-vaccinated mice and analyzing them via flow cytometry after ex vivo stimulation with BCG.
R: Please define this when appropriate.
Author’s response:
The part of the discussion about TLR2 and innate immune training was removed from the manuscript, according to the reviewer’s comments. Therefore, the cite about Jeyanathan et al. (2022) was also removed from the text.
Minor changes:
- L28 “of Toll-like receptor (TLR)-1, TLR2…”, same for IFN, iNOS, and CD – and please define the first time in the text
Author’s response:
Agreed.
- L31 takes > studied
Author’s response:
Agreed.
- L33 immunoreactivity > immunolabelling
Author’s response:
Agreed.
- L34 H-score – please define for the first time
Author’s response:
Agreed.
- L32 all lymphocytic? I think this is a typo
Author’s response:
Diffuse paucibacillary lesions are also called diffuse lymphocytic, since the inflammatory infiltrate is composed mainly of lymphocyte (Gonzalez et al., 2005). In the present study, one animal (which was not vaccinated) showed diffuse paucibacillary (lymphocytic) lesions. In the manuscript, we use both terms (lymphocytic and paucibacillary), to refer to this lesion type, which can be confusing. Therefore, L32 was modified to “… and diffuse paucibacillary (lymphocytic)”. Similar modifications were included in the Material and Methods section to highlight the fact that diffuse paucibacillary and lymphocytic lesions represent the same lesion type.
- L32 immunolabelled TLR2
Author’s response:
Agreed.
- L36 immunoreactivity for > immunolabelling of
Author’s response:
Agreed.
- L40 in the intestine
Author’s response:
Agreed.
- L46 chronic wasting disease is a prionic disease of deer, please find another way to briefly define PTB… chronic enteritis? Up to the authors
Author’s response:
Agreed.
- L82-83: Cancel “The tissue samples used in this study came from animals used in an old study for 82 other purposes [7].” Start the sentence as “Archived formalin-fixed…”
Author’s response:
Agreed.
- L85: Cancel “(ethical approval not specified)” and just add “[7]”
Author’s response:
Agreed.
- L82-92: Repeated, cancel
Author’s response:
Agreed.
- L92 Briefly, those samples selected > Selected samples were categorized
Author’s response:
Agreed.
- L490-503: Fuse
Author’s response:
Agreed.
- L497 dpv should be DPV as it is an abbreviation – correct along the text
Author’s response:
Agreed.
- L518-521 please cancel, this is obvious
Author’s response:
Agreed.
- L518, 522, 551: lymphocytes/neutrophils immunolabelling IFN – you are not addressing production
Author’s response:
Agreed.
- L522: On the other hand – Cancel
Author’s response:
Agreed.
- L524: “In fact” – Cancel
Author’s response:
Agreed.
- L535: immunoreactivity > immunolabelling
Author’s response:
Agreed.
- L536: In the present work > Here
Author’s response:
Agreed.
- L540: signs > paratuberculosis
Author’s response:
Agreed.
- L540: Taken together – Cancel
Author’s response:
Agreed.
- L544-545: In this sense, a weak > Early deficient innate…
Author’s response:
Agreed.
- L551 or phagocytosis…
Author’s response:
Agreed.
- Other: “n” should be italicized
Author’s response:
Agreed.
REFERENCES
González, J., Geijo, M. V., García-Pariente, C., Verna, A., Corpa, J. M., Reyes, L. E., Ferreras, M. C., Juste, R. A., García Marín, J. F., & Pérez, V. (2005). Histopathological classification of lesions associated with natural paratuberculosis infection in cattle. Journal of comparative pathology, 133(2-3), 184–196.
Zapico, D., Espinosa, J., Criado, M., Gutiérrez, D., Ferreras, M. D. C., Benavides, J., Pérez, V., & Fernández, M. (2025). Immunohistochemical expression of TLR1, TLR2, TLR4, and TLR9 in the different types of lesions associated with bovine paratuberculosis. Veterinary pathology, 62(3), 305–318.